# REPRESENTATIVE ACTION SELECTION FOR LARGE ACTION SPACE META-BANDITS

## ABSTRACT

We study the problem of selecting a subset from a large action space shared by a family of bandits, with the goal of achieving performance nearly matching that of using the full action space. We assume that similar actions tend to have related payoffs, modeled by a Gaussian process. To exploit this structure, we propose a simple $\epsilon$-net algorithm to select a representative subset. We provide theoretical guarantees for its performance and compare it empirically to Thompson Sampling and Upper Confidence Bound.

## 1 INTRODUCTION

We study a family of bandits that share a common but extremely large action space. We aim to understand whether it is possible—and how—to select a smaller set of representative actions that performs nearly as well as the full action space across all bandit instances. To build intuition, imagine a pharmacy preparing its inventory for the upcoming season. The available drugs (actions) are nearly infinite, and each customer (bandit) has unique characteristics. If two drugs share similar ingredients, their effects on a patient are likely to be similar. Likewise, if two patients have comparable health indices, a drug is likely to have similar effects on both. By modeling the expected outcome of each drug for each patient as a Gaussian process, we can capture these correlations. This correlation structure makes subset selection particularly interesting: if two drugs treat the same disease, choosing one may suffice, while drugs for distinct diseases might both be kept. In addition, demand also matters: drugs for rare illnesses can be excluded to save space, while flu medications should be stocked during flu season.

Different from prior approaches in multi-armed bandits (MAB) (Lai & Robbins, 1985) that aim for identifying either a single best action or a subset that achieves high cumulative outcomes for a fixed bandit, our objective focuses on selecting a subset that is likely to contain the best action, or one whose best element performs nearly as well for a family of bandits. This problem can be seen as a large-scale combinatorial optimization under uncertainty, with applications where decisions involve a vast number of possibilities but are constrained by computational or time limitations for evaluating all options, e.g., inventory management, online recommendations.

Consider the following setting: In a bandit, if a decision-maker plays an action with a fixed but unknown feature vector $a \in \mathcal{A}_{\text{full}} \subset \mathbb{R}^n$, they observe a random outcome taking values in $\mathbb{R}$. We define the expected outcome of playing action $a$ in this bandit as $\mu_a(\theta) := \langle a, \theta \rangle$ where this bandit instance $\theta \in \mathbb{R}^n$ is drawn from an unknown multivariate Gaussian distribution. Thus, the collection of random variables $\{\mu_a\}_{a \in \mathcal{A}_{\text{full}}}$ forms a Gaussian process (GP) (Vershynin, 2018, Chapter 7), while a more general sub-Gaussian assumption is considered in the Supplementary. Our setting aligns with that of contextual bandits (Dani et al., 2008), and the restriction to linear functions may be partially mitigated by allowing $n$ to be unbounded.

Consider that the decision-maker has access to a fixed action subset $\mathcal{A} \subset \mathcal{A}_{\text{full}}$. For a given bandit instance $\theta$, if the optimal action over the full space lies within $\mathcal{A}$, the decision-maker benefits from reduced suboptimal actions to explore. Conversely, if the optimal action lies outside $\mathcal{A}$, regret arises from being unable to select this best action. Thus, for a bandit instance $\theta$, we define the regret as the difference in expected outcome between having access to the full action space versus being restricted to the subset:

$$\text{Regret}(\theta) := \max_{a \in \mathcal{A}_{\text{full}}} \mu_a(\theta) - \max_{a' \in \mathcal{A}} \mu_{a'}(\theta), \tag{1}$$

which depends on the sampled bandit instance and is therefore a random variable. Our objective is to identify a small subset $\mathcal{A}$ that minimizes the expected regret $\mathbb{E}_\theta[\text{Regret}]$ over all possible bandit instances, making the underlying optimization both stochastic and combinatorial.

This objective is motivated by practical considerations. The classic Bayesian regret in the bandit literature (Agrawal & Goyal, 2012) typically scales with the number of available actions. However, if a subset $\mathcal{A}$ is carefully chosen, the resulting Bayesian bandit regret can be significantly lower. This is especially beneficial when the action space is large, as even the initialization phase can be computationally expensive. To see this, we can decompose the bandit regret as follows:

$$\text{BayesianBanditRegret} := \mathbb{E} \sum_{t=1}^{N} \left[ \max_{a \in \mathcal{A}_{\text{full}}} \mu_a(\theta) - \mu_{A_t}(\theta) \right]$$

$$= \mathbb{E} \sum_{t=1}^{N} \left[ \max_{a \in \mathcal{A}} \mu_a(\theta) - \mu_{A_t} + \max_{a \in \mathcal{A}_{\text{full}}} \mu_a(\theta) - \max_{a \in \mathcal{A}} \mu_a(\theta) \right]$$

$$\leq C\sqrt{|\mathcal{A}| \cdot N \log N} + N \cdot \mathbb{E}_\theta[\text{Regret}].$$

In the first equality, $A_t \in \mathcal{A}$ denotes the action chosen by a policy, e.g., Thompson Sampling (Agrawal & Goyal, 2012), in round $t$, and $N$ is the number of rounds. The expectation is taken over the randomness in the distribution of bandit instances and in actions selected by the policy. The inequality follows from the well-known regret bounds for Thompson Sampling (Lattimore & Szepesvári, 2020)[1], where $|\mathcal{A}|$ denotes the cardinality of the action set and $C > 0$ is a constant. Note that if the policy has access to the full action space, the Bayesian bandit regret is instead bounded by $C\sqrt{|\mathcal{A}_{\text{full}}| \cdot N \log N}$. Our main contributions are organized as follows:

- **Meta-Bandits Framework.** We propose a meta-bandits framework that specifically tackles combinatorial action selection by leveraging correlations across similar actions and bandit instances. To the best of our knowledge, this is the first such framework.

- $\epsilon$**-Net Algorithm.** We introduce a simple algorithm within this framework. It starts with the intuitive idea of placing a grid over the action space, then refines it using an importance-based selection mechanism.

- **Regret Analysis.** We provide theoretical guarantees for both the grid and the algorithm's output, including upper and lower bounds on expected regret, along with results under general sub-Gaussian processes. We also discuss the cost of not using a grid, which depends on the importance-structure of the action space.

- **Generalization and Empirical Validation.** We extend the analysis to settings where outcome functions are sampled from a reproducing kernel Hilbert space (RKHS), and empirically compare our algorithm to Thompson Sampling (TS) and Upper Confidence Bound (UCB).

## 1.1 RELATED WORKS

**Multi-Armed Bandits** (Lai & Robbins, 1985; Auer et al., 2002a) is defined by a set of actions (arms), each deliver outcomes that are independently drawn from a fixed and unknown distribution. The decision-maker sequentially selects an action, observes its outcome, and aims to maximize cumulative outcomes over time. Popular methods include the UCB (Auer et al., 2002a), TS (Agrawal & Goyal, 2012), and EXP3 (Auer et al., 2002b) for adversarial settings.

**Optimal Action Identification** focuses on identifying the action with the highest expected outcome in a MAB setting using as few samples as possible (Jamieson & Nowak, 2014; Kaufmann et al., 2016). Popular methods in fixed confidence setting include Action Elimination Even-Dar et al. (2006); Karnin et al. (2013), UCB, and LUCB, all of which achieve sample complexity within a $\log(|\mathcal{A}_{\text{full}}|)$ factor of the optimum. In fixed budget setting, there is Successive halving Karnin et al. (2013), successive reject Audibert & Bubeck (2010).

**Stochastic Linear Optimization** assumes that the expected outcome of each action depends through the inner product between a context $\theta$ and an action $a \in \mathcal{A}_{\text{full}}$ Auer (2002); Dani et al. (2008);

---

[1]This bound could be relaxed if the feature vector $a$ is known, or if $a$ can be incorporated as the input of a kernel function, as discussed in related works. However, we use this general bound because we also consider the most general case where $\mathcal{A}_{\text{full}}$ is simply an index set, e.g., $\mathcal{A}_{\text{full}} := \{\text{treatment A, treatment B, treatment C}\}$.

Rusmevichientong & Tsitsiklis (2010). This line of work assumes that the action feature vectors are known, so the cardinality of the action space does not play a role and may even be infinite. Soare et al. (2014) study the sample complexity of optimal action identification in this setting, which scales linearly with the action space dimension $n$.

**GP Optimization** addresses the case where the feature vectors of actions are unknown but a kernel function is available, so the inputs to the kernel are required. Srinivas et al. (2009) model the outcome function as a sample from a GP prior with a kernel function (Williams & Rasmussen, 2006), achieving bandit regret that scales as $\sqrt{\log|\mathcal{A}_{\text{full}}|}$ for finite action space. The widespread adoption of this method in bandit settings (Valko et al., 2013; Li & Scarlett, 2022), as well as in continuous action spaces (Chowdhury & Gopalan, 2017), highlights the practicality of assuming that action outcomes are correlated. This also motivates our extension, where the outcome function of each bandit instance is modeled as a sample from a RKHS.

Before discussing two seemingly relevant lines of work that select a subset from the action space, we first highlight an advantage of our approach. These methods assume a fixed subset cardinality $|\mathcal{A}| = K$, which is often unclear in practice and requires restarting the algorithm when changed. In contrast, our algorithm can adapt the subset size on the fly.

**Top-K Action Identification** aims to identify the $K$ actions with the highest expected outcomes using as few samples as possible (Kalyanakrishnan et al., 2012; Gabillon et al., 2012; Kaufmann et al., 2016; Chen et al., 2017). This line of work assumes that all actions are independent and have distinct expected values, making its methods inapplicable to our framework. If one were to apply these methods regardless, the most reasonable approach, in our view, would be to treat the family of bandit instances as a super-bandit, where each bandit instance corresponds to a round, and the expected payoff of an action in that round is given by $\mu_a$. In this setting, top-$K$ identification would refer to selecting $K$ actions with the highest expected payoffs $\mathbb{E}[\mu_a]$. In contrast, our framework considers that the expected outcome of each action, averaged over the distribution of bandits, may be the same—i.e., $\mathbb{E}[\mu_a] = c$ for all $a \in \mathcal{A}_{\text{full}}$, where $c$ is a constant—so that the entire action space shares the same highest expected payoff. Further, even if $\mathbb{E}[\mu_a]$ varies across actions, ignoring correlations can be fatal in our framework:

**Example 1.** *Consider three actions: $a_1 = [1, 0]$, $a_2 = [0.9, 0.1]$, and $a_3 = [-0.1, 1]$, and suppose bandits are sampled uniformly from $\theta_1 = [1, 0]$ and $\theta_2 = [0, 1]$. Then,*

$$\mathbb{E}\langle a_1, \theta \rangle = \mathbb{E}\langle a_2, \theta \rangle = 0.5, \quad \mathbb{E}\langle a_3, \theta \rangle = 0.45.$$

*So under the Best-2-Action perspective, $a_1$ and $a_2$ would be selected. However, this is suboptimal in our framework, since $\mu_{a_1}$ and $\mu_{a_2}$ are positively correlated:*

$$\mathbb{E} \max_{a \in \{a_1, a_2\}} \langle a, \theta \rangle = 0.55, \quad \mathbb{E} \max_{a \in \{a_1, a_3\}} \langle a, \theta \rangle = 1.$$

*Our algorithm, if run until it selects two distinct actions, would output $a_1$ and $a_3$—the true optimal.*

**Combinatorial Bandits** considers that the decision maker selects $K$ of base arms from $\mathcal{A}_{\text{full}}$ in each round, forming a super arm $\mathcal{A}$, with $|\mathcal{A}| = K$. Popular methods include CUCB Chen et al. (2016), CTS Wang & Chen (2018). We argue that this line of work is not directly applicable to our framework, but we include it as a baseline in subsequent empirical evaluations: (1) It assumes that the expected outcome of a super arm depends only on the expected outcomes of its individual base arms, or imposes a stricter monotonicity condition. In our case, even though $\mathbb{E}[\mu_a] = c$ for all $a \in \mathcal{A}_{\text{full}}$, super arm expected outcomes $\mathbb{E}[\max_{a \in \mathcal{A}} \mu_a]$ can differ significantly due to correlations among actions. (2) It assumes independence across base arms, whereas we explicitly model correlations. Ignoring these correlations misses the core challenge—an issue illustrated in Example 1.

## 2 SUBSET SELECTION FRAMEWORK

We consider the problem of selecting a small number of representative actions from a large action space $\mathcal{A}_{\text{full}} \subset \mathbb{R}^n$, where $n \in \mathbb{N}$. (This framework applies to the case $n = +\infty$, with the additional assumption $\sum_{i \geq 1} a_i^2 < \infty$.) The expected outcome depends on both the chosen action $a \in \mathbb{R}^n$ and an observed context $g \in \mathbb{R}^n$, and includes a constant $c \in \mathbb{R}$:

$$\mu_a := \langle a, g \rangle + c, \quad \forall\, a \in \mathcal{A}_{\text{full}},\ g \sim \mathcal{N}(0, \Sigma), \tag{2}$$

where $\Sigma$ is a positive semi-definite matrix. Let $\theta \in \mathbb{R}^n$ follows a multivariate normal distribution with zero mean and identity covariance matrix $I$. The distribution of $g$ is in fact equivalent to $\Sigma^{1/2}\theta$. Now, let $\sigma_j$ denote the $j$-row of the matrix $\Sigma^{1/2}$. We have $g = (\langle \sigma_j, \theta \rangle)_{j \leq n}$ and

$$\langle a, g \rangle = \sum_{j \leq n} a_j \langle \sigma_j, \theta \rangle = \left\langle \sum_{j \leq n} a_j \sigma_j, \theta \right\rangle = \langle \Sigma^{1/2} a, \theta \rangle.$$

Therefore, the setting in equation 2 is equivalent to $\mu_a := \langle a, \theta \rangle + c, \ \forall \ a \in \Sigma^{1/2}\mathcal{A}_{\text{full}}, \ \theta \sim \mathcal{N}(0, I)$, where $\Sigma^{1/2}\mathcal{A}_{\text{full}}$ denotes the image of $\mathcal{A}_{\text{full}}$ under the linear transformation $\Sigma^{1/2}$. Since the constant $c$ does not affect the regret (as defined in equation 1), we can, without loss of generality, focus on this canonical Gaussian process (Vershynin, 2018, Chapter 7) in the remainder:

$$\boxed{\mu_a(\theta) := \langle a, \theta \rangle, \quad \forall a \in \mathcal{A}_{\text{full}}, \theta \sim \mathcal{N}(0, I).} \tag{3}$$

We define the extreme points as those $x \in \mathcal{A}_{\text{full}}$ for which there do not exist distinct $a, a' \in \mathcal{A}_{\text{full}}$ and $\lambda \in (0, 1)$ such that $x = \lambda a + (1 - \lambda)a'$. By the extreme point theorem, if we select all extreme points—denoted $\mathcal{A} = \{a_1, \ldots, a_K\}$—as representatives of the full action space, the regret is zero. This is because any $a \in \mathcal{A}_{\text{full}}$ can be expressed as a convex combination of the extreme points: $a = \lambda_1 a_1 + \cdots + \lambda_K a_K$, where $\lambda_i \geq 0$ and $\sum_{i=1}^K \lambda_i = 1$. Thus, for any $\theta \in \mathbb{R}^n$:

$$\langle a, \theta \rangle = \sum_{i=1}^K \lambda_i \langle a_i, \theta \rangle \leq \sum_{i=1}^K \lambda_i \max_{a' \in \mathcal{A}} \langle a', \theta \rangle = \max_{a' \in \mathcal{A}} \mu_{a'}.$$

By equation 3, it yields $\max_{a' \in \mathcal{A}} \mu_{a'} \leq \max_{a \in \mathcal{A}_{\text{full}}} \mu_a \leq \max_{a' \in \mathcal{A}} \mu_{a'}$, where the left inequality uses $\mathcal{A} \subseteq \mathcal{A}_{\text{full}}$. Thus, the two quantities $\max_{a \in \mathcal{A}} \mu_a$ and $\max_{a \in \mathcal{A}_{\text{full}}} \mu_a$ are equal.

This example highlights how a geometric approach can be used to solve the stochastic combination problem. However, even if one only needs the extreme points, the set of extreme points may still be large, e.g, the extreme points of a Euclidean ball is infinite. To address this, we will later introduce the notions of $\epsilon$-nets. Without loss of generality, we assume $\mathcal{A}_{\text{full}}$ consists only of the extreme points of $\mathcal{A}_{\text{full}}$, as they are the only points of interest.

## 2.1 EPSILON NETS

If $\mathcal{A}_{\text{full}}$ is very large, a natural approach is to construct a grid over the action space, where the grid points serve as representative actions. This ensures that for every action in the full space, there exists a representative that is close to it. This idea is formally captured by the notion of a (geometric) $\epsilon$-net.

To proceed, we clarify what we mean by an $\epsilon$-net, as there are at least two definitions: one from a geometric perspective (Vershynin, 2018, Chapter 4) and another from a measure-theoretic perspective (Matousek, 2013, Chapter 10). Let $\|\cdot\|_2$ denote the Euclidean norm. Define the diameter of a compact set $r \in \mathbb{R}^n$ as $\text{diam}(r) := \max_{a,b \in r} \|a - b\|_2$.

- A subset $\mathcal{A} \subseteq \mathcal{A}_{\text{full}}$ is called a **Geometric $\epsilon$-net** if, for all $a \in \mathcal{A}_{\text{full}}$, there exists $a' \in \mathcal{A}$ such that
$$\|a - a'\|_2 < \epsilon.$$

- Let $\mathcal{R}$ be a finite partition of the extreme points into disjoint clusters such that $\cup_{r \in \mathcal{R}} \ r = \mathcal{A}_{\text{full}}$. Given a measure $q$ assigning a value to each cluster $r \in \mathcal{R}$. A subset $\mathcal{A} \subseteq \mathcal{A}_{\text{full}}$ is called a **Measure-Theoretic $\epsilon$-net** with respect to measure $q$ if, for any cluster $r \in \mathcal{R}$, we have:
$$r \cap \mathcal{A} \neq \emptyset \quad \text{whenever} \quad q(r) > \epsilon.$$

A geometric $\epsilon$-net ensures small regret because if two actions $a, a' \in \mathcal{A}_{\text{full}}$ are close in the Euclidean sense, then the deviation between $\mu_a$ and $\mu_{a'}$ is small in the $L^2$-sense (i.e., their expected squared difference is small): $\|\mu_a - \mu_{a'}\|_{L^2} = \left(\mathbb{E}(a - a')^\top \theta \theta^\top (a - a')\right)^{1/2} = \|a - a'\|_2$, where $\theta^\top$ denotes the transpose of $\theta$. The equalities use equation 3 and $\mathbb{E}\theta\theta^\top = I$. Therefore, by definition, a geometric $\epsilon$-net guarantees the existence of an action $a \in \mathcal{A}$ whose expected outcome $\mu_a$ is close to that of the optimal action for any given bandit instance. However, this net suffers from the curse of dimension: e.g., for $[0, 1]^n$, the number of points needed to form a geometric $\epsilon$-net grows as $(1/\epsilon)^n$.

The measure-theoretic $\epsilon$-net addresses this issue. Put simply, the measure-theoretic $\epsilon$-net restricts the grid construction to only the most important clusters $r \in \mathcal{R}$, as determined by the $q$-measure.

---

**Algorithm 1** Epsilon Net Algorithm

---
1: **Input:** Action space $\mathcal{A}_{\text{full}}$, Sample size $K$.
2: **Output:** A subset of actions $\mathcal{A}$.
3: $\mathcal{A} \leftarrow \emptyset$
4: **for** $1, \ldots, K$ **do**
5:    Sample a bandit instance $\theta$.
6:    Find optimal action $a^*(\theta) := \arg\max_{a \in \mathcal{A}_{\text{full}}} \langle a, \theta \rangle$.
7:    $\mathcal{A} \leftarrow \mathcal{A} \cup \{a^*\}$                  ▷ Repetition of actions is allowed
8: **end for**

---

## 2.2 EPSILON NET ALGORITHM

We propose Algorithm 1, a variant of the $\epsilon$-net algorithm originally introduced by Haussler & Welzl (1986). It selects $K$ i.i.d. random actions, aligned with the distribution of bandit instances. Since repetitions are allowed, the resulting subset $\mathcal{A}$ may have fewer than $K$ distinct actions.

We define the optimal action in a bandit instance $\theta$ as

$$a^*(\theta) := \arg\max_{a \in \mathcal{A}_{\text{full}}} \mu_a(\theta).$$

**Assumption 2** (Unique optimal action)**.** *The optimal action $a^*(\theta)$ is unique with probability $1$ over all bandit instances.*

Define the **Importance Measure** $q$ over a partition $\mathcal{R}$:

$$\boxed{q(r) := \Pr[a^*(\theta) \in r] = \int \mathbb{1}\{a^*(\theta) \in r\}\, p(\theta)d\theta,} \tag{4}$$

where $p(\theta)$ is the density of $\theta$ and $\int p(\theta)d\theta = 1$. Under Assumption 2, measure $q$ is a probability distribution. It reflects the probability that a given cluster contains the optimal action and thus represents the potential contribution of that cluster to the expected regret.

**Assumption 3.** *The support of measure $q$ is compact.*

The compactness assumption ensures that the term $\mathbb{E}_\theta\left[\max_{a \in \mathcal{A}_{\text{full}}} \mu_a\right]$ is finite and guarantees the attainment of a unique optimal action. Without loss of generality, we assume that $\mathcal{A}_{\text{full}}$ is the support of the measure $q$.

By definition, Algorithm 1 samples $K$ i.i.d. extreme points from clusters in $\mathcal{R}$ according to measure $q$. If a cluster $r \in \mathcal{R}$ has a higher measure $q(r)$, its elements are more likely to be included in the output. In fact, with high probability, this algorithm outputs a measure-theoretic $\epsilon$-net of $\mathcal{A}_{\text{full}}$ with respect to measure $q$. (This is a simplified version of Theorem 10.2.4 of Matousek (2013).)

**Lemma 4.** *Given a partition $\mathcal{R}$ of the full action space, and the importance measure $q$ assigning a value to each cluster $r \in \mathcal{R}$. Let $\mathcal{A}$ be the output of Algorithm 1 after $K$ samples. Then, with probability at least $1 - \frac{1}{\epsilon}\exp(-K\epsilon)$, it holds that for any cluster $r \in \mathcal{R}$,*

$$r \cap \mathcal{A} \neq \emptyset \quad \text{whenever} \quad q(r) > \epsilon.$$

The partition $\mathcal{R}$ bridges the two definitions of $\epsilon$-net: choosing one point from each cluster gives an $\epsilon$-net in both senses, though with different values of $\epsilon$. Geometrically, $\epsilon$ is the largest cluster diameter $\epsilon := \max_{r \in \mathcal{R}} \text{diam}(r)$; measure-theoretically, $\epsilon$ is the smallest cluster measure $\epsilon := \min_{r \in \mathcal{R}} q(r)$.

## 3 REGRET ANALYSIS

In this section, we begin by analyzing a special class of geometric $\epsilon$-nets, constructed by partitioning the action space into clusters and selecting a single representative action from each cluster. We then extend the analysis to obtain algorithm-dependent bounds for the output of Algorithm 1.

**Definition 1** (Reference subsets)**.** *Consider a partition $\mathcal{R} := \{r_\ell\}_{\ell \leq m}$ of the full action space, with $\epsilon := \max_{r \in \mathcal{R}} \text{diam}(r)$. A reference subset is a set $\mathcal{A} := \{a_1, \ldots, a_m\}$, where each representative $a_\ell \in \mathcal{A}$ corresponds to a cluster $r_\ell$ and each cluster $r_\ell$ is contained within a closed Euclidean ball of radius $\epsilon$ centered at $a_\ell$, i.e., $r_\ell \subset B(a_\ell, \epsilon)$.*

Only for lower bounds, we assume a well-separated partition: if the optimal action lies in cluster $r$, then all actions in $r$ outperform those outside it.

**Assumption 5.** *For any $r \in \mathcal{R}$, whenever the optimal action lies in cluster $r$, i.e., $a^*(\theta) \in r$, then $\mu_a \geq \max_{a' \in \mathcal{A}_{\text{full}} \setminus r} \mu_{a'}, \ \forall a \in r$.*

**Theorem 6** (Regret bounds of reference subsets). *Consider a partition $\mathcal{R} := \{r_\ell\}_{\ell \leq m}$ of the full action space, with $\epsilon := \max_{r \in \mathcal{R}} \operatorname{diam}(r)$, and an arbitrary reference subset $\mathcal{A}$. Then, there is an absolute constant $C > 0$, such that*

$$\mathbb{E}_\theta[\text{Regret}] \leq \max_{r \in \mathcal{R}} \mathbb{E}_\theta\left[\max_{a \in r} \mu_a\right] + C\epsilon\sqrt{\log|\mathcal{R}|}.$$

*If the partition $\mathcal{R}$ satisfies Assumption 5, then*

$$\mathbb{E}_\theta[\text{Regret}] \geq \min_{r \in \mathcal{R}} \mathbb{E}_\theta\left[\max_{a \in r} \mu_a\right] - C\epsilon\sqrt{\log|\mathcal{R}|}.$$

*Proof sketch.* For each cluster $\ell \leq m$, define a simple Gaussian process $\{Z_a\}_{a \in r_\ell}$ where $Z_a := \mu_a - \mu_{a_\ell}$. Define a non-negative random variable $Y_\ell := \sup_{a \in r_\ell} Z_a$. When $a^*(\theta) \in r_\ell$, regret is upper bounded by $Y_\ell$ (or equal to it under Assumption 5). Thus, for any bandit $\theta$, the regret is bounded between $\min_{\ell \leq m} Y_\ell$ and $\max_{\ell \leq m} Y_\ell$. Finally, the expectations $\mathbb{E}[\min_{\ell \leq m} Y_\ell]$ and $\mathbb{E}[\max_{\ell \leq m} Y_\ell]$ can be bounded via concentration property of Gaussian process. □

For each $\ell \leq m$, the term $\mathbb{E}_\theta[\max_{a \in r_\ell} \mu_a]$ equals $\mathbb{E}_\theta[\max_{a \in r_\ell} \mu_a - \mu_{a_\ell}]$, since $\mathbb{E}_\theta[\mu_a] = 0$.

## 3.1 Regret Bounds of Algorithm

The algorithm's regret bound is established by comparing it to that of a reference subset, for which we already have known expected regret bounds. The key difference is that whereas the reference subset includes a representative from each cluster, the algorithm may miss some clusters. However, the algorithm still achieves regret comparable to that of the reference subset, as it tends to miss clusters that contribute minimally to the expected regret.

The expected regret in previous results is taken over bandit instances $\theta$. In contrast, since the output of Algorithm 1 is random, the expected regret analyzed in this section is taken with respect to both the algorithm's randomness (i.e., the sampled $\mathcal{A}$) and the distribution over $\theta$.

**Theorem 7.** *Consider a partition $\mathcal{R} := \{r_\ell\}_{\ell \leq m}$ of the action space, with $\epsilon := \max_{r \in \mathcal{R}} \operatorname{diam}(r)$. Let $\mathcal{A}$ be output of Algorithm 1. For the same constant $C > 0$ in Theorem 6,*

$$\mathbb{E}_{\theta,\mathcal{A}}[\text{Regret}] \leq \max_{r \in \mathcal{R}} \mathbb{E}_\theta\left[\max_{a \in r} \mu_a\right] + C\epsilon\sqrt{\log|\mathcal{R}|} + \left(\mathbb{E}_q\left[(1 - q(r))^{2K}\right] \cdot \mathbb{E}_\theta\left[\max_{a \in \mathcal{A}_{\text{full}}} \mu_a^2\right]\right)^{1/2}.$$

*Proof sketch.* If $r_\ell \cap \mathcal{A} \neq \emptyset$, choose $a_\ell \in r_\ell \cap \mathcal{A}$ as the representative point. If $r_\ell \cap \mathcal{A} = \emptyset$, choose an arbitrary point $a_\ell \in r_\ell$. The new set $\mathcal{A}' := \{a_\ell\}_{\ell \leq m}$ forms a reference subset. Then, for each $\ell \leq m$, define a Gaussian process $\{Z_a\}_{a \in r_\ell}$, where $Z_a := \mu_a - \mu_{a_\ell}$, and let $Y_\ell := \sup_{a \in r_\ell} Z_a$. When $a^*(\theta) \in r_\ell$, we consider two cases. If $r_\ell \cap \mathcal{A} \neq \emptyset$, the regret is upper bounded by $Y_\ell$, and hence by $\max_{\ell \leq m} Y_\ell$. If $r_\ell \cap \mathcal{A} = \emptyset$, the regret is bounded by $\max_{a \in \mathcal{A}_{\text{full}}} \mu_a$. □

The term $\mathbb{E}_q\left[(1 - q(r))^{2K}\right]$ in Theorem 7 is maximized by the uniform $q$, provided $|\mathcal{R}| \geq 2K + 1$ (since we can choose any partition); see Supplementary. Intuitively, each cluster $r$ contributes $q(r) \cdot (1 - q(r))^{2K}$. This expression is small when $q(r)$ is small, and decays faster when $q(r)$ gets larger, making the overall term negligible if $q$ is highly concentrated.

The connection between Theorem 6 and 7 is insightful:

**Remark 8.** *Consider placing a grid over the action space and defining a partition $\mathcal{R}$ by assigning each point in the space to its nearest grid point. In this way, the grid acts as a reference subset with respect to the partition $\mathcal{R}$. The regret upper bound in Theorem 6 applies to this grid, as it holds for any reference subset. Meanwhile, Theorem 7 applies to any partition, including $\mathcal{R}$. As a result, we obtain a regret bound for Algorithm 1 that exceeds the grid's bound by only one additional term:*

$$\left(\mathbb{E}_q\left[(1 - q(r))^{2K}\right] \cdot \mathbb{E}_\theta\left[\max_{a \in \mathcal{A}_{\text{full}}} \mu_a^2\right]\right)^{1/2},$$

*where the part $\mathbb{E}_\theta \left[ \max_{a \in \mathcal{A}_{\text{full}}} \mu_a^2 \right]$ is simply a constant, as the action space $\mathcal{A}_{\text{full}}$ is fixed.*

*If the distribution $q$ is highly concentrated, then the expectation $\mathbb{E}_q[(1 - q(r))^{2K}]$ is small, making the extra term negligible. In this case, the benefit of constructing an explicit grid—often a non-trivial task when the action feature vectors are unknown or high-dimensional—is limited.*

Generally speaking, the partition $\mathcal{R}$ may be unknown, and while the corresponding $q$ measure exists, it remains unspecified. We therefore provide a worst-case bound using the *covering number* $N(\mathcal{A}_{\text{full}}, \epsilon)$, which is the smallest number of points needed to form a geometric $\epsilon$-net of $\mathcal{A}_{\text{full}}$:

$$N(\mathcal{A}_{\text{full}}, \epsilon) = \min \left\{ m \in \mathbb{N} : \; \exists \{a_\ell\}_{\ell \leq m} \subseteq \mathcal{A}_{\text{full}}, \forall a \in \mathcal{A}_{\text{full}}, \exists \ell, \|a - a_\ell\|_2 \leq \epsilon \right\}.$$

**Theorem 9.** *Under Assumption 3, there exists a point $a_0$ and a constant $M > 0$ such that $\mathcal{A}_{\text{full}} \subset B(a_0, M)$, a closed Euclidean ball of radius $M$ centered at $a_0$. Let the action space have dimension $n$, and fix a constant $0 < \epsilon < M$. Let $\mathcal{A}$ be the output of Algorithm 1. For the same constant $C > 0$ in Theorem 6, and another absolute constant $c > 0$,*

$$\mathbb{E}_{\theta, \mathcal{A}}[\text{Regret}] \leq 2\epsilon\sqrt{n} + C\epsilon\sqrt{\log N(\mathcal{A}_{\text{full}}, \epsilon)}, \text{where } K \geq c \cdot (M/\epsilon)^2 \cdot N(\mathcal{A}_{\text{full}}, \epsilon).$$

*As $\epsilon \to 0^+$, we have $\mathbb{E}_{\theta, \mathcal{A}}[\text{Regret}] \to 0$ as $K \to \infty$.*

The partition-independent upper bound in Theorem 9 depends on a positive constant $\epsilon$, which serves as a tolerance parameter up to the diameter of the action space. Given a $\epsilon$, the required number of samples $K$ scales with the square of the diameter-to-$\epsilon$ ratio, multiplied by the covering number. The resulting expected regret is then bounded in terms of $\epsilon$, the dimensionality, and the logarithm of the covering number. As $\epsilon$ decreases, more samples are needed, but the regret bound becomes tighter.

The matching lower bound for Theorem 7, as well as the upper bound under the general sub-Gaussian process assumption for Algorithm 1, are provided in the Supplementary.

## 4 GENERALIZATION AND EMPIRICAL VALIDATION

In general cases, the decision-maker may not have explicit access to the full structure of the action space, especially in high-dimensional settings. Instead, they are given a list of actions and can observe the expected outcomes of these actions through sampling. We therefore treat the expected outcomes $\{\mu_a\}_{a \in \mathcal{A}_{\text{full}}}$ as a family of random variables indexed by an abstract set, e.g., $\mathcal{A}_{\text{full}} := \{\text{treatment A}, \text{treatment B}, \text{treatment C}\}$. In a single bandit, these expected outcomes define an **outcome function** $f$ over the action space $\mathcal{A}_{\text{full}}$: $f(a) := \mu_a, \forall a \in \mathcal{A}_{\text{full}}$. We show that our framework applies when the outcome functions lie in a reproducing kernel Hilbert space (RKHS) (Williams & Rasmussen, 2006). Consider a positive semidefinite kernel $k : \mathcal{A}_{\text{full}} \times \mathcal{A}_{\text{full}} \to \mathbb{R}$. By Mercer's theorem, for a non-negative measure $\mathbb{P}$ over $\mathcal{A}_{\text{full}}$, if the kernel satisfies $\int_{\mathcal{A}_{\text{full}} \times \mathcal{A}_{\text{full}}} k^2(a, a') d\mathbb{P}(a) d\mathbb{P}(a') < \infty$, then it admits an eigenfunction expansion: $k(a, a') = \sum_{i \leq \infty} \lambda_i \phi_i(a) \phi_i(a')$, where $(\phi_i)_{i \leq \infty}$ are orthonormal eigenfunctions under $\mathbb{P}$, and $(\lambda_i)_{i \leq \infty}$ are non-negative eigenvalues. Let outcome function $f$ be of the form $f(\cdot) = \sum_{i=1}^N \alpha_i k(\cdot, a_i)$ for some integer $N \geq 1$, and a set of points $\{a_i\}_{i=1}^N \subset \mathcal{A}_{\text{full}}$ and a weight vector $\alpha \in \mathbb{R}^N$. The function $f$ can be rewritten as

$$\mu_a = f(a) = \langle \mathbf{f}, \Phi(a) \rangle,$$

where $\mathbf{f}$ is a vector of coefficients, and $\Phi(a)$ usually refereed as a feature map, has entries $\Phi_i(a) = \sqrt{\lambda_i}\phi_i(a)$. The full action space is the one formed by the feature vectors $\Phi(a)$ for $a \in \mathcal{A}_{\text{full}}$.

Assuming the outcome function is generated from a kernel inherently implies that actions are correlated. To study the effect of varying correlation levels among actions, we use the parametric RBF kernel to sample outcome functions from a RKHS: $k(a, a') := \exp\left( -\frac{\|a - a'\|^2}{2l^2} \right)$, where $l$ is a length-scale parameter that control the dependence among actions. By (Kanagawa et al., 2018, Theorem 4.12), we sample functions from a RKHS by first constructing the kernel matrix $\mathbf{K}$ with entries $\mathbf{K}_{a,a'} = k(a, a')$ for $a, a' \in \mathcal{A}_{\text{full}}$, and then drawing $f \sim \mathcal{N}(0, \mathbf{K})$. Note that although the kernel is used to generate outcome functions, the kernel itself is not revealed; consequently, $a$ and $a'$ are regarded merely as indices rather than as inputs to the kernel.

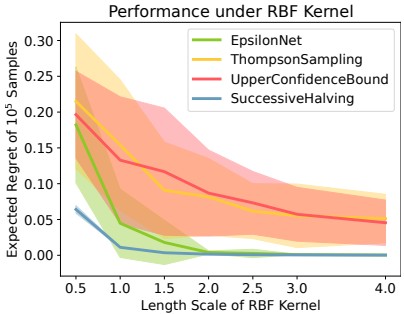
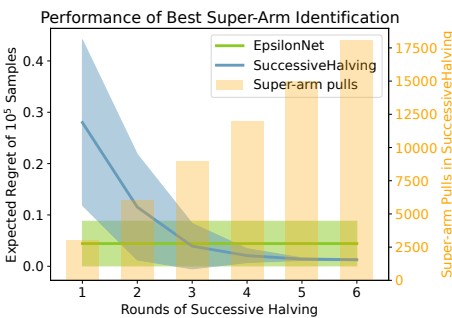

Figure 1: Comparison on solving equation 5, selecting $K = 5$ distinct actions from 15 grid points over $[0, 2]$, with outcome functions $f(a) := \mu_a$ sampled from an RBF kernel at varying length-scales. **Left:** Expected regret over 50 repetitions of our method with exhaustive search (green), compared with Thompson Sampling (TS; yellow), Upper Confidence Interval (UCB; orange), and Successive Halving (SH, blue) applied to the set of super-arms. TS/UCB are run for 3000 rounds; SH is given a budget of 37,000 pulls. **Right:** SH's expected regret (blue) and super-arm pull counts (yellow bars) per round, compared with our method (green). SH requires nearly 10,000 pulls—about three times the number of super-arms—to match our method.

Returning to the objective stated in the Introduction, we aim to find a subset $\mathcal{A}$ of cardinality $K$ that minimizes the expected regret defined in equation 1. Since the term $\mathbb{E}[\max_{a \in \mathcal{A}_{\text{full}}} \mu_a]$ is a constant independent of $\mathcal{A}$, this is equivalent to the following optimization problem:

$$\max_{\mathcal{A} \subseteq \mathcal{A}_{\text{full}}} \mathbb{E}\left[\max_{a \in \mathcal{A}} \mu_a\right] \quad \text{subject to } |\mathcal{A}| = K. \tag{5}$$

In general, the problem is non-trivial since the objective involves an expectation over random variables with an unspecified distribution, and the inner expression is a maximum over a subset of potentially correlated variables. Consequently, small changes in $\mathcal{A}$ can cause non-smooth variations in the maximum. Our method is computationally efficient, avoiding the combinatorial complexity.

We assess performance in identifying near-optimal solutions by selecting $K = 5$ actions from an action space of 15 grid points in $[0, 2]$. The configuration yields a moderate number of super arms, allowing approximation of the optimal subset using Successive Halving (SH), which, however, becomes impractical for larger action spaces. Outcome functions are generated with an RBF kernel at length-scales $l \in \{0.5, 1, \ldots, 4\}$. We compare the proposed method against Thompson Sampling (TS) and Upper Confidence Bound (UCB). In our method, we run Algorithm 1 until $K = 5$ distinct actions are selected, where exhaustive search is used over the action space to find the optimal action $a^*(\theta)$. For other methods, we treat each $K$-tuple of actions as a super arm, and the payoff of each super arm in a round $\mathcal{A}$ is given by $\max_{a \in \mathcal{A}} \mu_a$. Thus, these methods are tailored to find the best super arm that maximizes $\mathbb{E}[\max_{a \in \mathcal{A}} \mu_a]$, aligning with the same objective. In TS/UCB methods, we adopt a bandit feedback setting: at each round (corresponding to a bandit instance $\theta$), the decision-maker selects a super arm $\mathcal{A}$, observes the payoff $\max_{a \in \mathcal{A}} \mu_a(\theta)$, and updates its policy accordingly, repeating this process for 3000 rounds, chosen to roughly match the number of super arms ($N = 3003$). Since the payoff $\max_{a \in \mathcal{A}} \mu_a$ is unbounded, we assume Gaussian payoffs for both TS and UCB, with the prior set to $\mathcal{N}(0, 1)$. In the SH method, all super arms are evaluated using a fixed budget of arm pulls (we use the minimum budget required in Karnin et al. (2013), $N \log_2 N \approx 37,000$) over a few rounds. In each round (corresponding to a bandit instance $\theta$), the number of remaining super arms is halved. This process continues until only the best-performing super arm remains or the budget is exhausted. We evaluate expected regret of all methods over $10^5$ additional randomly sampled outcome functions.

The left subplot of Figure 1 reports the expected regret of our method (green), TS (yellow), UCB (orange) and SH (blue), where solid curves and error shades indicate the mean $\pm$ one standard deviation of expected regret over 50 repetitions. As shown, the expected regret decreases as the length-scale increases. The length-scale $l$ controls the effective number of approximately independent actions: when $a$ and $a'$ are far relative to $l$, $k(a, a')$ is negligible, so $\mu_a$ and $\mu_{a'}$ are nearly uncorrelated. In the limit, $\{\mu_a\}_{a \in \mathcal{A}_{\text{full}}}$ forms a collection of i.i.d. zero-mean, unit-variance variables.

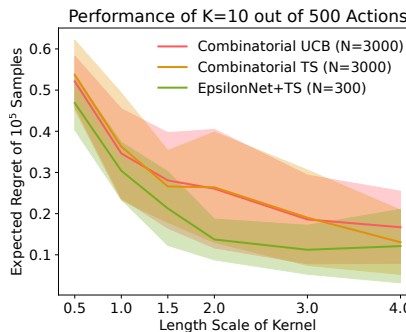

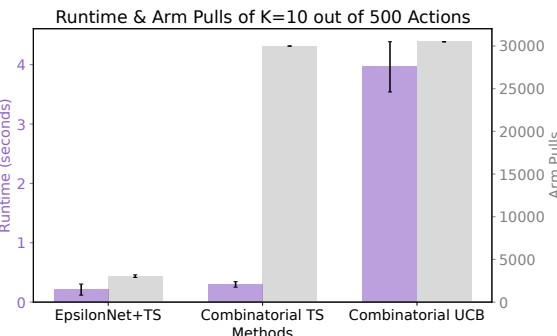

Figure 2: Comparison of our method (green), CTS (brown), and CUCB (red) on solving equation 5 with $K = 10$ actions chosen from 500 grid points over $[-5, 5]$, with outcome functions $f(a) := \mu_a$ sampled from an RBF kernel at varying length-scales. In our method, EpsilonNet+TS, the "argmax" step in Algorithm 1 is replaced by TS run for 300 rounds. CTS and CUCB are run for 3000 rounds. All methods use prior variance 1. **Left:** expected regret over 30 repetitions. **Right:** runtime (purple, left axis) and number of arm pulls (gray, right axis) over 30 repetitions.

The SH in left subplot of Figure 1 is used to approximate the ground-truth optimal subset and our method does give similar performance as SH. The right subplot highlights SH's impracticality: we ran our method (green) and SH (blue) for 50 repetitions with fixed $l = 1$, tracking the current best super arm and total super arm pulls after each SH round. The blue curves and shades represent the mean $\pm$ one standard deviation of SH' expected regret, while yellow bars (corresponding to the right y-axis) show the number of super arm pulls by SH at the end of each round. Green curves show our method's expected regret. SH matches our performance after three rounds but requires 9005 pulls—roughly three times the number of super arms—making it intractable for large $\mathcal{A}_{\text{full}}$.

Further, we demonstrate the generality of our method (denoted EpsilonNet+TS), where the "argmax" step in Algorithm 1 is replaced by TS. We consider the same setup as in Figure 1, but with $K = 10$ actions chosen from 500 grid points over $[-5, 5]$. In our method, for each bandit instance $\theta$, we run TS for 300 rounds, selecting an action $a$ and observing $\mu_a$ in each round; the final action is taken as an approximation of the optimal action $a^*(\theta)$. We repeat this process until $K = 10$ distinct actions are identified. We compare against two heuristics: combinatorial TS (CTS) and combinatorial UCB (CUCB). CTS and CUCB are run for 3000 rounds. Both CTS and CUCB operate in the semi-bandit feedback setting: in each round (corresponding to one bandit instance $\theta$), an outcome function is sampled and $K = 10$ actions $\mathcal{A}$ are selected, with the rewards $\mu_a(\theta)$ revealed for all $a \in \mathcal{A}$. The prior for TS, CTS, and CUCB is set to $\mathcal{N}(0, 1)$, consistent with the ground truth.

In Figure 2, the left subplot reports the expected regret and its standard deviation for EpsilonNet+TS (green), CTS (brown), and CUCB (red), averaged over 30 repetitions. The poor performance of CTS and CUCB is expected, as noted in related works. The variance of EpsilonNet+TS could be further reduced by running more TS rounds or using a more accurate action-identification method, since we currently use a Gaussian prior—modeling each action's expected outcome $\mu_a$ as an independent Gaussian random variable—rather than a Gaussian process prior, in order to accommodate the most general case where $\mathcal{A}_{\text{full}}$ is an index set. In the right subplot, we fix the length-scale $l = 1$ and evaluate runtime and arm pulls of the three methods over 30 repetitions. The left axis (purple) shows average runtime $\pm$ one standard deviation, and the right axis (gray) shows the corresponding number of arm pulls. EpsilonNet+TS requires 3000 pulls, since it selects $K = 10$ optimal actions, with each action identified by running TS for 300 rounds. CTS has similar runtime but needs far more pulls, as it runs 3000 rounds with $K = 10$ arms pulled per round. CUCB runs much slower, and the additional arm pulls result from the extra initialization step it requires.

**Conclusion and Future Work** We proposed a framework for selecting a subset of correlated actions. A simple algorithm was introduced and shown to effectively identify near-optimal subsets. Future work could develop a stopping criterion. When subset size is flexible but sampling new bandit instances is costly, ideas from species discovery (Roswell et al., 2021) may be useful—treating the discovery of a new action as analogous to discovering a new species.

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

**Appendix Summary:** The appendix is divided into four parts.

First, Sections A-B provide additional related works and the regret bound of Algorithm 1, which were omitted from the main body due to page limits.

Second, Sections C to H contain the proofs of all the theorems and lemmas stated both in the main body and in Section B.

Third, Sections I-J present auxiliary tools used throughout the paper.

Fourth, Sections K-M include additional examples and a numerical study of Algorithm 1. This part features a low-dimensional ($n = 3$) action space on the unit sphere to investigate the effect of cluster diameter on expected regret, in line with Theorem 6. It also studies two extensions of the varying dependence structure of actions introduced in the main body:

- Nonstationary dependence over the action space: Using a nonstationary Gibbs kernel, where the kernel value depends not only on the difference between two inputs $a, a'$ but also on the location of $a$.
- Limiting case: The collection of expected outcomes $\{\mu_a\}_{a \in \mathcal{A}_{\text{full}}}$ becomes i.i.d.

**LLM Usage:** Large language models are used for polishing writing, and finding relevant research.

## A  EXTRA RELATED WORKS

**Epsilon Nets** have two standard definitions. The first, geometric definition (Vershynin, 2018), requires that radius-$\epsilon$ balls centered at net points cover the set. It relates to the covering number and extends to function classes, as in Russo & Van Roy (2013). The second, measure-theoretic definition (Matousek, 2013), requires the net to intersect all subsets of sufficiently large measure. The classic $\epsilon$-net algorithm by Haussler & Welzl (1986) remains the simplest and most broadly applicable method. Later works aim to reduce net size (Pach & Tardos, 2011; Rabani & Shpilka, 2009; Mustafa, 2019) and address online settings (Bhore et al., 2024).

**Expected supremum of Gaussian process** for a given set $\mathcal{S}$ refers to the term $\mathbb{E}\left[\max_{a \in \mathcal{S}} \mu_a\right]$. It is an important topic in high-dimensional probability (Vershynin, 2018). The sharpest known bounds are due to Talagrand (2014). and asymptotic convex geometry (Rothvoss, 2021, Chapter 9)

## B  EXTRA REGRET BOUNDS OF ALGORITHM 1

We give a matching lower bound:

**Theorem 10.** *Under the same condition of Theorem 7. Let each cluster contains more than one action and satisfies Assumption 5. For the same constant $C$ in Theorem 6 and another absolute constant $c > 0$,*

$$\mathbb{E}_{\theta, \mathcal{A}}[\text{Regret}] \geq \left( \min_{r \in \mathcal{R}} \mathbb{E}_\theta \left[ \max_{a \in r} \mu_a \right] - C\epsilon \sqrt{\log |\mathcal{R}|} \right) \times c \cdot \left( \mathbb{E}_q \left[ (1 - q(r))^{2K} \right] \right)^{1/2}.$$

In fact, the Gaussian process assumption can be relaxed to a sub-Gaussian one, where the random process $\{\mu_a\}_{a \in \mathcal{S}}$ satisfies the following increment condition:

$$\forall u > 0, \quad \Pr\left[ |\mu_a - \mu_{a'}| \geq u \right] \leq 2 \exp\left( -\frac{u^2}{2\|a - a'\|_2^2} \right).$$

Let $\gamma_2(\mathcal{S})$ denote Talagrand's chaining functional (Talagrand, 2014) for a set $\mathcal{S} \subset \mathbb{R}^n$, $n \in \mathbb{N}$, equipped with the Euclidean norm. It is the sharpest known bound that for a sub-Gaussian process indexed by $\mathcal{S}$, there exists a constant $c > 0$ such that $\mathbb{E}\left[\max_{a \in \mathcal{S}} \mu_a\right] \leq c\gamma_2(\mathcal{S})$. Using this functional, we establish a regret bound for our algorithm:

**Theorem 11.** *Let $\{\mu_a\}_{a \in \mathcal{A}_{\text{full}}}$ be a mean-zero sub-Gaussian process. Consider a partition $\mathcal{R} := \{r_\ell\}_{\ell \leq m}$ of the full action space. Let $\mathcal{A}$ be output of Algorithm 1. Then, for a constant $C > 0$,*

$$\mathbb{E}_{\theta, \mathcal{A}}[\text{Regret}] \leq C\sqrt{\log m} \cdot \max_{\ell \leq m} \gamma_2(r_\ell) + \left( \mathbb{E}_q \left[ (1 - q(r))^{2K} \right] \cdot \mathbb{E}_\theta \left[ \max_{a \in \mathcal{A}_{\text{full}}} \mu_a^2 \right] \right)^{1/2}.$$

Proofs are in Appendices G and H, respectively.

## C  PROOF OF LEMMA 4

**Statement:** Given a partition $\mathcal{R}$ of the full action space, and the importance measure $q$ assigning a value to each cluster $r \in \mathcal{R}$. Let $\mathcal{A}$ be the output of Algorithm 1 after $K$ samples. Then, with probability at least $1 - \frac{1}{\epsilon}\exp(-K\epsilon)$, it holds that for any cluster $r \in \mathcal{R}$,

$$r \cap \mathcal{A} \neq \emptyset \quad \text{whenever} \quad q(r) > \epsilon.$$

*Proof.* By definition, the algorithm draws $K$ i.i.d. samples from clusters in $\mathcal{R}$ according to the distribution $q$. Define a set of typical clusters $R_\epsilon := \{r \in \mathcal{R} : q(r) > \epsilon\}$. Then,

$$\Pr[r \cap \mathcal{A} = \emptyset] = (1 - q(r))^K < (1 - \epsilon)^K \leq \exp(-K\epsilon), \ \forall r \in R_\epsilon, \qquad (6)$$

where the equality is the probability of missing the cluster $r$ for $K$ times, the first inequality uses the definition of $R_\epsilon$. The last inequality uses $\log(1-\epsilon)^K = K\log(1-\epsilon)$ and the inequality $\log(1-\epsilon) \leq -\epsilon$ for $\epsilon < 1$ (TOPSØE[1], 2007). Therefore,

$$\Pr[r \cap \mathcal{A} \neq \emptyset \text{ whenever } q(r) > \epsilon]$$
$$= 1 - \Pr[r \cap \mathcal{A} \neq \emptyset \ \exists r \in R_\epsilon]$$
$$\geq 1 - \sum_{r \in R_\epsilon} \Pr[r \cap \mathcal{A} \neq \emptyset]$$
$$> 1 - \sum_{r \in R_\epsilon} \exp(-K\epsilon)$$
$$= 1 - |R_\epsilon|\exp(-K\epsilon) \geq 1 - 1/\epsilon \cdot \exp(-K\epsilon),$$

where the first inequality uses union bound, the second inequality uses equation 6. The last inequality uses the fact that there could be at most $\lfloor 1/\epsilon \rfloor$ clusters in $R_\epsilon$. $\qquad \square$

## D  PROOF OF THEOREM 6

Under Assumption 5, the regret can be decomposed as follows; note that this Lemma is used only to derive lower bounds.

**Lemma 12** (Regret decomposition of reference subsets). *Consider a partition $\mathcal{R} := \{r_\ell\}_{\ell \leq m}$ of the full action space and an arbitrary reference subset $\mathcal{A}$. For each $\ell \leq m$, define a Gaussian process $\{Z_a\}_{a \in r_\ell}$ where $Z_a := \mu_a - \mu_{a_\ell}$. Define a non-negative random variable*

$$Y_\ell := \sup_{a \in r_\ell} Z_a = \sup_{a \in r_\ell} \mu_a - \mu_{a_\ell}.$$

*Under Assumption 5,* Regret $= Y_\ell$ *where the cluster $r_\ell$ contains the optimal action. Hence,*

$$\mathbb{E}_\theta[\text{Regret}] = \sum_{\ell \leq m} q(r_\ell) \cdot \mathbb{E}_\theta\left[Y_\ell \middle| a^*(\theta) \in r_\ell\right].$$

*Proof.* Let the cluster $r_\ell$ contains the optimal action, i.e., $a^*(\theta) \in r_\ell$, then

$$\text{Regret} = \max_{a \in \mathcal{A}_{\text{full}}} \mu_a - \max_{a' \in \mathcal{A}} \mu_{a'} = \max_{a \in r_\ell} \mu_a - \mu_{a_\ell} = Y_\ell,$$

where the middle equality uses that when $a^*(\theta) \in r_\ell$, $\max_{a \in \mathcal{A}_{\text{full}}} \mu_a = \max_{a \in r_\ell} \mu_a$ and further with Assumption 5, $\max_{a' \in \mathcal{A}} \mu_{a'} \geq \mu_{a_\ell}$ when $a^*(\theta) \in r_\ell$. Then,

$$\mathbb{E}[\text{Regret}] = \sum_{\ell \leq m} \Pr[a^*(\theta) \in r_\ell] \cdot \mathbb{E}\left[Y_\ell \middle| a^*(\theta) \in r_\ell\right] = \sum_{\ell \leq m} q(r_\ell) \cdot \mathbb{E}\left[Y_\ell \middle| a^*(\theta) \in r_\ell\right],$$

where the left equality uses tower rule, the right equality uses the definition of $q$ measure. $\qquad \square$

To bound the expected regret of reference subsets, we need a few more lemmas:

**Lemma 13** (Expectation integral identity). *Given a non-negative random variables $X$. If $\Pr[X \geq u] \leq c\exp\left(-\frac{u^2}{\epsilon^2}\right)$ for any $u > 0$, then $\mathbb{E}X \leq C\epsilon\sqrt{\log c}$, where $\epsilon, c, C$ are positive constants.*

*Proof.*

$$\mathbb{E}X = \int_0^\infty \Pr[X \ge u] \, du$$

$$= \int_0^{u_0} \Pr[X \ge u] \, du + \int_{u_0}^\infty \Pr[X \ge u] \, du$$

$$\le u_0 + \frac{1}{u_0} \int_{u_0}^\infty u \cdot \Pr[X \ge u] \, du$$

$$\le u_0 + \frac{c}{u_0} \int_{u_0}^\infty u \cdot \exp\left(-\frac{u^2}{\epsilon^2}\right) \, du$$

$$= u_0 + \exp\left(-\frac{u_0^2}{\epsilon^2}\right) \cdot \frac{c\epsilon^2}{2u_0}$$

$$= \epsilon\sqrt{\log c} + \frac{\epsilon}{2\sqrt{\log c}} \le C\epsilon\sqrt{\log c},$$

where the first equality uses integrated tail formula of expectation (cf. Lemma 1.6.1 of Vershynin (2018)). The last equality set $u_0 := \epsilon\sqrt{\log c}$. $\qquad\square$

**Lemma 14** (Borell-TIS inequality; Lemma 2.4.7 of (Talagrand, 2014))**.** *Given a set $\mathcal{S}$, and a zero-mean Gaussian process $(X_a)_{a \in \mathcal{S}}$. Let $\epsilon := \sup_{a \in \mathcal{S}} \left(\mathbb{E}X_a^2\right)^{\frac{1}{2}}$. Then for $u > 0$, we have*

$$\Pr\left[\left|\sup_{a \in \mathcal{S}} X_a - \mathbb{E}\sup_{a \in \mathcal{S}} X_a\right| \ge u\right] \le 2\exp\left(-\frac{u^2}{2\epsilon^2}\right).$$

It means that the size of the fluctuations of $\mathbb{E}\sup_{a \in \mathcal{S}} X_a$ is governed by the size of the individual random variables $X_a$.

**Statement of Theorem 6:** Consider a partition $\mathcal{R} := \{r_\ell\}_{\ell \le m}$ of the full action space, with $\epsilon := \max_{r \in \mathcal{R}} \text{diam}(r)$, and an arbitrary reference subset $\mathcal{A}$. Then, for some constant $C > 0$,

$$\mathbb{E}_\theta[\text{Regret}] \le \max_{r \in \mathcal{R}} \mathbb{E}_\theta\left[\max_{a \in r} \mu_a\right] + C\epsilon\sqrt{\log|\mathcal{R}|}.$$

If the partition $\mathcal{R}$ satisfies Assumption 5, then

$$\mathbb{E}_\theta[\text{Regret}] \ge \min_{r \in \mathcal{R}} \mathbb{E}_\theta\left[\max_{a \in r} \mu_a\right] - C\epsilon\sqrt{\log|\mathcal{R}|}.$$

*Proof.* Fix $\ell$, define a Gaussian process $\{Z_a\}_{a \in r_\ell}$ where $Z_a := \mu_a - \mu_{a_\ell}$. Define a non-negative random variable

$$Y_\ell := \sup_{a \in r_\ell} Z_a = \sup_{a \in r_\ell} \mu_a - \mu_{a_\ell}.$$

Since $\mathbb{E}\mu_a = 0$ for all $a \in \mathcal{A}_{\text{full}}$, it holds that $\mathbb{E}Y_\ell = \mathbb{E}\max_{a \in r_\ell} \mu_a$.

When the cluster $r_\ell$ contains the optimal action, i.e., $a^*(\theta) \in r_\ell$, we have

$$\text{Regret} = \max_{a \in \mathcal{A}_{\text{full}}} \mu_a - \max_{a' \in \mathcal{A}} \mu_{a'} = \max_{a \in r_\ell} \mu_a - \max_{a' \in \mathcal{A}} \mu_{a'} \le \max_{a \in r_\ell} \mu_a - \mu_{a_\ell} = Y_\ell \le \max_{\ell' \le m} Y_{\ell'},$$

where the first equality follows from the definition of regret in equation 1, and the second equality follows from the assumption that $a^*(\theta) \in r_\ell$. The inequality holds because $a_\ell \in \mathcal{A}$, and hence $\max_{a \in \mathcal{A}} \mu_a \ge \mu_{a_\ell}$. The final equality follows from the definition of $Y_\ell$. On the other hand, according to Lemma 12, under Assumption 5, $\text{Regret} = Y_\ell$, and thus $\text{Regret} \ge \min_{\ell' \le m} Y_{\ell'}$, where the cluster $r_\ell$ contains the optimal action.

Therefore, we reach the important conclusion that

$$\min_{\ell \le m} Y_\ell \le \text{Regret} \le \max_{\ell \le m} Y_\ell, \tag{7}$$

where the left inequality holds under Assumption 5. Further, by definition $r_\ell \subset B(a_\ell, \epsilon)$, such that $\mathbb{E}Z_a^2 = \mathbb{E}(\mu_a - \mu_{a_\ell})^2 = \|a - a_\ell\|_2^2 \leq \epsilon^2$. Using Lemma 14 on the process $\{Z_a\}_{a \in r_\ell}$, we have

$$\Pr\left[|Y_\ell - \mathbb{E}Y_\ell| \geq u\right] \leq 2\exp\left(-\frac{u^2}{2\epsilon^2}\right).$$

By union bound, we have

$$\Pr\left[\max_{\ell \leq m} |Y_\ell - \mathbb{E}Y_\ell| \geq u\right] \leq 2m\exp\left(-\frac{u^2}{2\epsilon^2}\right).$$

Using Lemma 13, we have for some absolute constant $C > 0$

$$\mathbb{E}\max_{\ell \leq m} |Y_\ell - \mathbb{E}Y_\ell| \leq C\epsilon\sqrt{\log m}. \tag{8}$$

**Upper bound:**

$$\mathbb{E}\,\mathrm{Regret} \leq \mathbb{E}\max_{\ell \leq m} Y_\ell \leq \max_{\ell \leq m} \mathbb{E}Y_\ell + \mathbb{E}\max_{\ell \leq m} |Y_\ell - \mathbb{E}Y_\ell| \leq \max_{\ell \leq m} \mathbb{E}\max_{a \in r_\ell} \mu_a + C\epsilon\sqrt{\log m}, \tag{9}$$

where the first inequality uses equation 7. The second inequality uses $\max_{\ell \leq m} Y_\ell \leq \max_{\ell \leq m} \mathbb{E}Y_\ell + \max_{\ell \leq m} |Y_\ell - \mathbb{E}Y_\ell|$, since $Y_\ell \leq \mathbb{E}Y_\ell + |Y_\ell - \mathbb{E}Y_\ell|$. The last inequality uses equation 8 and the identity $\mathbb{E}[Y_\ell] = \mathbb{E}[\max_{a \in r_\ell} \mu_a]$.

**Lower bound:**

$$\mathbb{E}\,\mathrm{Regret} \geq \mathbb{E}\min_{\ell \leq m} Y_\ell \geq \min_{\ell \leq m} \mathbb{E}Y_\ell - \mathbb{E}\max_{\ell \leq m} |Y_\ell - \mathbb{E}Y_\ell| \geq \min_{\ell \leq m} \mathbb{E}\max_{a \in r_\ell} \mu_a - C\epsilon\sqrt{\log m}, \tag{10}$$

where the first inequality uses equation 7 under Assumption 5. The second inequality uses $\min_{\ell \leq m} Y_\ell \geq \min_{\ell \leq m} \mathbb{E}Y_\ell - \max_{\ell \leq m} |Y_\ell - \mathbb{E}Y_\ell|$, since $Y_\ell \geq \mathbb{E}Y_\ell - |Y_\ell - \mathbb{E}Y_\ell|$. The last inequality uses equation 8 and the identity $\mathbb{E}[Y_\ell] = \mathbb{E}[\max_{a \in r_\ell} \mu_a]$. $\qquad\square$

# E   PROOF OF THEOREM 7

An important consequence of assuming $\mathbb{E}_\theta[\mu_a] = 0$ for all $a \in \mathcal{A}_{\mathrm{full}}$ is the following property:

**Lemma 15** (Transformation invariance)**.** *Given any vector $x \in \mathbb{R}^n$, let $S + x := \{s + x : s \in \mathcal{S}\}$. If $\mathbb{E}[\theta] = 0$, then $\mathbb{E}[\max_{a \in \mathcal{S}+x}\langle a, \theta\rangle] = \mathbb{E}[\max_{a' \in \mathcal{S}}\langle a', \theta\rangle]$.*

*Proof.*

$$\mathbb{E}\left[\max_{a \in \mathcal{S}+x}\langle a, \theta\rangle\right] = \mathbb{E}\left[\max_{a' \in \mathcal{S}}\langle a' + x, \theta\rangle\right]$$
$$= \mathbb{E}\left[\max_{a' \in \mathcal{S}}\langle a', \theta\rangle\right] + \mathbb{E}\left[\langle x, \theta\rangle\right]$$
$$= \mathbb{E}\left[\max_{a' \in \mathcal{S}}\langle a', \theta\rangle\right],$$

where the last equality uses $\mathbb{E}[\langle x, \theta\rangle] = \langle x, \mathbb{E}[\theta]\rangle = 0$. $\qquad\square$

The key tool for decomposing the regret of Algorithm 1 into that from a reference subset and that from missing clusters is the following Lemma:

**Lemma 16.** *Consider a partition $\mathcal{R}$ of the full action space. Let $\mathcal{A}$ be the output of Algorithm 1. Then, the event that the optimal action falls in a cluster $r$, i.e., $\{a^*(\theta) \in r\}$, is independent of whether the subset $\mathcal{A}$ intersects with the cluster. It holds that*

$$\Pr[a^*(\theta) \in r, r \cap \mathcal{A} = \emptyset] = q(r)(1 - q(r))^K,$$
$$\Pr[a^*(\theta) \in r, r \cap \mathcal{A} \neq \emptyset] \leq q(r).$$

*Proof.* Since $\mathcal{A}$ is the output of Algorithm 1, the event $\{a^*(\theta) \in r\}$ is independent from $\{r \cap \mathcal{A} \neq \emptyset\}$ or $\{r \cap \mathcal{A} = \emptyset\}$. We have

$$
\begin{aligned}
\Pr[a^*(\theta) \in r, r \cap \mathcal{A} = \emptyset] &= \Pr[a^*(\theta) \in r] \Pr[r \cap \mathcal{A} = \emptyset] \\
&= q(r) \Pr[r \cap \mathcal{A} = \emptyset] \\
&= q(r)(1 - q(r))^K,
\end{aligned}
$$

where the first equality uses independence between $\{a^*(\theta) \in r\}$ and $\{r \cap \mathcal{A} \neq \emptyset\}$, the second equality uses the definition of measure $q$, the third equality use the probability of missing cluster $r$ in $K$ i.i.d. samplings. Similarly,

$$
\Pr[a^*(\theta) \in r, r \cap \mathcal{A} \neq \emptyset] = q(r) \cdot \Pr[r \cap \mathcal{A} \neq \emptyset].
$$

Using $\Pr[r \cap \mathcal{A} \neq \emptyset] \leq 1$, we complete the proof. $\qquad\square$

**Statement of Theorem 7:** Let $\mathcal{A}$ be output of Algorithm 1. Consider a partition $\mathcal{R} := \{r_\ell\}_{\ell \leq m}$ of the full action space, with $\epsilon := \max_{r \in \mathcal{R}} \mathrm{diam}(r)$. Then, for the same constant $C > 0$ in Theorem 6,

$$
\begin{aligned}
\mathbb{E}_{\theta, \mathcal{A}}[\text{Regret}] \leq &\max_{r \in \mathcal{R}} \mathbb{E}_\theta \left[ \max_{a \in r} \mu_a \right] + C\epsilon\sqrt{\log |\mathcal{R}|} \\
&+ \left( \mathbb{E}_q \left[ (1 - q(r))^{2K} \right] \cdot \mathbb{E}_\theta \left[ \max_{a \in \mathcal{A}_{\text{full}}} \mu_a^2 \right] \right)^{1/2}.
\end{aligned}
$$

*Proof.* If $r_\ell \cap \mathcal{A} \neq \emptyset$, define $a_\ell \in r_\ell \cap \mathcal{A}$. If $r_\ell \cap \mathcal{A} = \emptyset$, choose an arbitrary point $a_\ell \in r_\ell$ as the representative. The set $\mathcal{A}' := \{a_\ell\}_{\ell \leq m}$ forms a reference subset of Definition 1. The cluster $r_\ell$ is contained in a closed Euclidean ball of radius $\epsilon$ centered at $a_\ell$, i.e., $r_\ell \subset B(a_\ell, \epsilon)$. Define a Gaussian process $\{Z_a\}_{a \in r_\ell}$ where $Z_a := \mu_a - \mu_{a_\ell}$. Define a random variable

$$
Y_\ell := \sup_{a \in r_\ell} Z_a = \sup_{a \in r_\ell} \mu_a - \mu_{a_\ell}.
$$

Since $\mathbb{E}\mu_a = 0$ for all $a \in \mathcal{A}_{\text{full}}$, we have $\mathbb{E}Y_\ell = \mathbb{E}\sup_{a \in r_\ell} \mu_a$.

Consider the case that $a^*(\theta) \in r_\ell$. We have

$$
\begin{aligned}
&\mathbb{E}\left[ \text{Regret} \,\Big|\, r_\ell \cap \mathcal{A} \neq \emptyset, a^*(\theta) \in r_\ell \right] \\
&\leq \mathbb{E}\left[ Y_\ell \,\Big|\, r_\ell \cap \mathcal{A} \neq \emptyset, a^*(\theta) \in r_\ell \right] \\
&= \mathbb{E}\left[ Y_\ell \,\Big|\, a^*(\theta) \in r_\ell \right], \\
&\leq \mathbb{E}\left[ \max_{\ell \leq m} Y_\ell \,\Big|\, a^*(\theta) \in r_\ell \right],
\end{aligned}
\tag{11}
$$

where the first inequality uses $\text{Regret} \leq Y_\ell$ if $r_\ell \cap \mathcal{A} \neq \emptyset$, the equality uses that $Y_\ell$ is independent of $r_\ell \cap \mathcal{A} \neq \emptyset$. On the other hand, we assume that $0 \in \mathrm{Conv}(\mathcal{A})$ because even if it doesn't hold, we can always find a vector $x \in \mathbb{R}^n$ such that $0 \in \mathrm{Conv}(\mathcal{A} + x)$, without changing the value of $\mathbb{E}[\max_{a \in \mathcal{A}} \mu_a]$ (c.f. Lemma 15). Therefore,

$$
\begin{aligned}
&\mathbb{E}\left[ \text{Regret} \,\Big|\, r_\ell \cap \mathcal{A} = \emptyset, a^*(\theta) \in r_\ell \right] \\
&\leq \mathbb{E}\left[ \max_{a \in \mathcal{A}_{\text{full}}} \mu_a \,\Big|\, a^*(\theta) \in r_\ell \right],
\end{aligned}
\tag{12}
$$

where the inequality uses Regret $\leq \max_{a \in \mathcal{A}_{\text{full}}} \mu_a$, as a consequence of $0 \in \text{Conv}(\mathcal{A})$, and that $\max_{a \in \mathcal{A}_{\text{full}}} \mu_a$ is independent of $r_\ell \cap \mathcal{A} = \emptyset$. Further,

$$
\begin{aligned}
\mathbb{E}[\text{Regret}] &= \sum_{r \in \mathcal{R}} \Pr[r \cap \mathcal{A} \neq \emptyset, a^*(\theta) \in r] \cdot \mathbb{E}\left[\text{Regret} \,\Big|\, r \cap \mathcal{A} \neq \emptyset, a^*(\theta) \in r\right] \\
&\quad + \sum_{r \in \mathcal{R}} \Pr[r \cap \mathcal{A} = \emptyset, a^*(\theta) \in r] \cdot \mathbb{E}\left[\text{Regret} \,\Big|\, r \cap \mathcal{A} = \emptyset, a^*(\theta) \in r\right] \\
&\leq \sum_{\ell \leq m} q(r_\ell) \cdot \left(\mathbb{E}\left[\max_{\ell \leq m} Y_\ell \,\Big|\, a^*(\theta) \in r_\ell\right] + (1 - q(r_\ell))^K \cdot \mathbb{E}\left[\max_{a \in \mathcal{A}_{\text{full}}} \mu_a \,\Big|\, a^*(\theta) \in r_\ell\right]\right) \\
&= \mathbb{E}\left[\max_{\ell \leq m} Y_\ell\right] + \sum_{r \in \mathcal{R}} q(r) \cdot (1 - q(r))^K \cdot \mathbb{E}\left[\max_{a \in \mathcal{A}_{\text{full}}} \mu_a \,\Big|\, a^*(\theta) \in r\right] \\
&\leq \mathbb{E}\left[\max_{\ell \leq m} Y_\ell\right] + \left(\mathbb{E}_q\left[(1 - q(r))^{2K}\right] \cdot \mathbb{E}_\theta\left[\max_{a \in \mathcal{A}_{\text{full}}} \mu_a^2\right]\right)^{1/2},
\end{aligned}
$$
(13)

where the first equality uses tower rule. The first inequality uses Equations (11-12), and Lemma 16. The last equality uses tower rule again. The last inequality uses Cauchy-Schwarz inequality, which states that for any two random variables $X$ and $Y$, we have $|\mathbb{E}[XY]| \leq \sqrt{\mathbb{E}[X^2]\mathbb{E}[Y^2]}$.

Note that the Gaussian process assumption is not used in equation 13; it is only needed to bound the term $\mathbb{E}\left[\max_{\ell \leq m} Y_\ell\right]$. By applying equation 9 in the proof of Theorem 6, we bound this term and complete the proof. $\qquad\square$

## F    PROOF OF THEOREM 9

To analyze the worst-case behavior of Theorem 7, we first examine the term $\mathbb{E}_q\left[(1 - q(r))^K\right]$ in the upper bound, summarized in the following Lemma:

**Lemma 17.** *Let $q$ denote a discrete probability distribution over a finite support $\mathcal{R}$. Define*

$$
M := \max_q \mathbb{E}_q\left[(1 - q(r))^K\right]
$$
$$
\text{s.t.} \sum_{r \in \mathcal{R}} q(r) = 1, \ q(r) \in [0, 1) \ \forall r \in \mathcal{R}.
$$
(14)

*When $|\mathcal{R}| \geq K + 1$, the maximum is $M = \left(1 - \frac{1}{|\mathcal{R}|}\right)^K$, and it is attained when $q$ is uniform. When $|\mathcal{R}| < K + 1$, the maximum is upper bounded by $M \leq \frac{|\mathcal{R}|}{K+1}\left(\frac{K}{K+1}\right)^K$, and there exists a feasible distribution $q'$ such that $\mathbb{E}_{q'}\left[(1 - q'(r))^K\right] \geq \frac{|\mathcal{R}|-1}{K+1}\left(\frac{K}{K+1}\right)^K$.*

*Proof.* Let $\mathcal{R} = \{r_1, \ldots, r_m\}$ be the finite support of the measure $q$, where $m := |\mathcal{R}|$. Let $q_i := q(r_i)$ denote the probability mass at each support point. Define the function $f(q_\ell) := q_\ell \cdot (1 - q_\ell)^K$.

**Case I** $m < K + 1$**:** The first derivative of $f$ is

$$
f'(q_\ell) = (1 - q_\ell)^{K-1}\big(1 - (K + 1)q_\ell\big),
$$

which is positive on the interval $\left[0, \frac{1}{K+1}\right)$ and negative on the interval $\left(\frac{1}{K+1}, 1\right)$. Therefore, $f(q_\ell)$ attains its maximum over $[0, 1)$ at

$$
q_\ell^* = \frac{1}{K + 1},
$$

with the corresponding maximum value

$$
f(q_\ell^*) \leq \frac{1}{K + 1}\left(\frac{K}{K + 1}\right)^K.
$$

Since there are $m$ support points, this yields the upper bound $M \leq \sum_{\ell=1}^{m} f(q_\ell^*) \leq \frac{m}{K+1} \left( \frac{K}{K+1} \right)^K$.

Also, since $\frac{1}{K+1} < \frac{1}{m}$, the solution $q_1 = \cdots = q_{m-1} = \frac{1}{K+1}$ and $q_m = \frac{K-m+2}{K+1}$ is feasible. Thus,

$$M \geq \sum_{\ell=1}^{m-1} f\left( \frac{1}{K+1} \right) + f\left( \frac{K-m+2}{K+1} \right)$$

$$\geq \frac{m-1}{K+1} \left( \frac{K}{K+1} \right)^K,$$

where the right inequality uses that $f(q_\ell) \geq 0$ for $q_\ell \in [0,1]$.

**Case II** $m \geq K+1$: Consider the relaxed maximization problem of equation 14:

$$\max_{q_1,\ldots,q_m \in [0,1)} \sum_{\ell=1}^{m} f(q_\ell) \quad \text{subject to} \quad \sum_{\ell=1}^{m} q_\ell \leq 1. \tag{15}$$

Let $\lambda \geq 0$ be the Lagrange multiplier associated with the constraint. Define the Lagrangian:

$$\mathcal{L}(q_1,\ldots,q_m,\lambda) = \sum_{\ell=1}^{m} q_\ell (1-q_\ell)^K - \lambda \left( \sum_{\ell=1}^{m} q_\ell - 1 \right).$$

For each $\ell = 1,\ldots,m$, compute the partial derivative of $\mathcal{L}$ with respect to $q_\ell$:

$$\frac{\partial}{\partial q_\ell} \left[ q_\ell (1-q_\ell)^K \right] = (1-q_\ell)^K - K q_\ell (1-q_\ell)^{K-1}.$$

Setting this derivative equal to zero yields the stationary condition:

$$(1-q_\ell)^{K-1}(1 - (K+1)q_\ell) = \lambda, \quad \text{with } \lambda \geq 0.$$

To find critical points of equation 15, we solve the system:

$$(1-q_\ell)^{K-1}(1 - (K+1)q_\ell) = \lambda \geq 0 \quad \forall \ell \leq m, \quad \text{and} \quad \sum_{\ell=1}^{m} q_\ell \leq 1.$$

Define the function $g(q_\ell) := (1-q_\ell)^{K-1}(1 - (K+1)q_\ell)$. For $g(q_\ell) \geq 0$, it must hold that $1 - (K+1)q_\ell \geq 0$, i.e., $q_\ell \leq \frac{1}{K+1}$. Therefore, any feasible solution to this system must satisfy $q_\ell \in \left[ 0, \frac{1}{K+1} \right]$ for all $\ell \leq m$.

Then, over the interval $q_\ell \in \left[ 0, \frac{1}{K+1} \right]$, both factors $(1-q_\ell)^{K-1}$ and $1 - (K+1)q_\ell$ are positive and decreasing. Hence, $g(q_\ell)$ is positive and strictly decreasing, so the equation $g(q_\ell) = \lambda \geq 0$ has at most one solution. Therefore, all $q_\ell$'s must be equal at a critical point. Let $q_\ell = c$ for all $\ell \leq m$. Due to the assumption of $m \geq K+1$, we have $\frac{1}{m} \leq \frac{1}{K+1}$, so any choice of $c \in \left[ 0, \frac{1}{m} \right]$ satisfies the constraint $\sum_{\ell=1}^{m} q_\ell \leq 1$ and is feasible for the system.

Therefore, $q_\ell = c$ for all $\ell \leq m$ is a feasible critical point of equation 15. The corresponding objective value is:

$$\sum_{\ell=1}^{m} f(c) = m \cdot c \cdot (1-c)^K,$$

which is increasing in $c$ over the interval $\left[ 0, \frac{1}{m} \right]$. Hence, the maximum is attained at $c = \frac{1}{m}$, and the optimal value is:

$$\sum_{\ell=1}^{m} f\left( \frac{1}{m} \right) = \left( 1 - \frac{1}{m} \right)^K = \left( 1 - \frac{1}{|\mathcal{R}|} \right)^K,$$

achieved when $q_\ell = \frac{1}{m}$ for all $\ell$.

To confirm that this critical point is indeed a maximum, observe that the Hessian of the objective function $f(q_\ell)$ is diagonal (since the function is separable), and the diagonal entries are:

$$\frac{\partial^2}{\partial q_\ell^2} \left[ q_\ell (1-q_\ell)^K \right] = -(1-q_\ell)^{K-2} \left( 2K - K(K+1)q_\ell \right),$$

which is negative for $q_\ell \leq \frac{1}{K+1}$, because then $2 - (K+1)q_\ell > 0$. Hence, the Hessian is negative definite, and the critical point is a local (and thus global) maximum.

Finally, note that equation 15 is a relaxation of equation 14. While the optimum of the original problem is upper bounded by that of the relaxed problem, the optimal solution to the relaxed problem also lies within the feasible region of the original problem. Therefore, the maximum of equation 14 is also attained when $q$ is uniform, with the maximum value being $\left(1 - \frac{1}{|\mathcal{R}|}\right)^K$. $\qquad\square$

**Statement of Theorem 9** Under Assumption 3, there exists a point $a_0$ and a constant $M > 0$ such that $\mathcal{A}_{\text{full}} \subset B(a_0, M)$, a closed Euclidean ball of radius $M$ centered at $a_0$. Let the action space have dimension $n \in \mathbb{N}$, and fix a constant $0 < \epsilon < M$. Let $\mathcal{A}$ be the output of Algorithm 1. For the same constant $C > 0$ in Theorem 6, we have:

$$\mathbb{E}_{\theta, \mathcal{A}}[\text{Regret}] \leq 2\epsilon\sqrt{n} + C\epsilon\sqrt{\log N(\mathcal{A}_{\text{full}}, \epsilon)}, \quad \text{where } K \geq \frac{1}{2}\left(\frac{M^2 N(\mathcal{A}_{\text{full}}, \epsilon)}{\epsilon^2 e} - 1\right).$$

As $\epsilon \to 0^+$, we have $\mathbb{E}_{\theta, \mathcal{A}}[\text{Regret}] \to 0$ as $K \to \infty$.

*Proof.* By Lemma 15, we can shift the action space to be centered at the origin. So, without loss of generality, we assume that $\mathcal{A}_{\text{full}} \subset M \cdot B_2^n$, the scaled unit Euclidean ball in $\mathbb{R}^n$. Then,

$$\mathbb{E}_\theta\left[\max_{a \in \mathcal{A}_{\text{full}}} \mu_a^2\right] \leq M^2 \cdot \mathbb{E}\|\theta\|_2^2 = M^2 n, \tag{16}$$

where the inequality follows from $\mu_a = \langle \theta, a \rangle \leq M\|\theta\|_2$ for all $a \in MB_2^n$, and the equality uses that $\theta \sim \mathcal{N}(0, I)$, so each coordinate has variance 1 and $\|\theta\|_2^2 = \sum_{i=1}^n \theta_i^2$ has expectation $n$.

Let $\{a_1, \ldots, a_m\} \subseteq \mathcal{A}_{\text{full}}$ be a minimal geometric $\epsilon$-net under the Euclidean norm, so that $m = N(\mathcal{A}_{\text{full}}, \epsilon)$. Define $\pi(a)$ as the closest point in the $\epsilon$-net to $a$, and let the partition $\mathcal{R} = \{r_1, \ldots, r_m\}$ be given by

$$r_\ell = \{a \in \mathcal{A}_{\text{full}} : \pi(a) = a_\ell\}.$$

Then,

$$\max_{\ell \leq m} \mathbb{E}_\theta\left[\max_{a \in r_\ell} \mu_a\right] \leq \mathbb{E}_\theta\left[\max_{a \in B(a_\ell, \epsilon)} \mu_a\right] = \mathbb{E}_\theta\left[\max_{a \in \epsilon B_2^n} \mu_a\right] \leq \epsilon\sqrt{n}, \tag{17}$$

where the first inequality follows from $r_\ell \subset B(a_\ell, \epsilon)$ by construction; the equality uses Lemma 15 to shift; and the final bound uses Claim 3 from Supplementary I. Then,

$$\mathbb{E}_{\theta, \mathcal{A}}[\text{Regret}] \leq \max_{r \in \mathcal{R}} \mathbb{E}_\theta\left[\max_{a \in r} \mu_a\right] + C\epsilon\sqrt{\log m} + \left(\mathbb{E}_q\left[(1 - q(r))^{2K}\right] \cdot \mathbb{E}_\theta\left[\max_{a \in \mathcal{A}_{\text{full}}} \mu_a^2\right]\right)^{1/2},$$

$$\leq \epsilon\sqrt{n} + C\epsilon\sqrt{\log m} + \sqrt{\frac{m}{2K+1}\left(\frac{2K}{2K+1}\right)^K} \cdot M\sqrt{n}$$

$$\leq \epsilon\sqrt{n} + C\epsilon\sqrt{\log m} + \epsilon\sqrt{n},$$

where the first inequality follows from the algorithm-dependent upper bound in Theorem 7. The second inequality follows from Equations equation 16 and equation 17, and the term $\mathbb{E}_q\left[(1 - q(r))^{2K}\right]$ is bounded by $\frac{m}{2K+1}\left(\frac{2K}{2K+1}\right)^{2K}$ (see Lemma 17). The last inequality comes from

$$\log\left(\sqrt{\frac{m}{2K+1}\left(\frac{2K}{2K+1}\right)^K} M\right) = \frac{1}{2}\log\left(\frac{m}{2K+1}\right) + K\log\left(\frac{2K}{2K+1}\right) + \log M$$

$$\leq \frac{1}{2}\log m - \frac{1}{2}\log(2K+1) - \frac{1}{2} + \log M$$

$$\leq \frac{1}{2}\log m - \frac{1}{2}\log\left(\frac{mM^2}{\epsilon^2 e}\right) - \frac{1}{2} + \log M \leq \log \epsilon,$$

where the first inequality uses $\frac{1}{2}\log\left(\frac{m}{2K+1}\right) = \frac{1}{2}(\log m - \log(2K+1))$. Also, $\log\left(\frac{2K}{2K+1}\right) = \log\left(1 - \frac{1}{2K+1}\right) \leq -\frac{1}{2K+1}$ due to the inequality $\log(1 - X) \leq -X$ for $X < 1$, and for a large $K$, we approximate $-\frac{K}{2K+1} \approx -\frac{1}{2}$. The last inequality uses the assumption $K \geq \frac{1}{2}\left(\frac{mM^2}{\epsilon^2 e} - 1\right)$.

Finally, since $\mathcal{A}_{\text{full}} \subset MB_2^n$, the covering number satisfies $N(\mathcal{A}_{\text{full}}, \epsilon) \leq C'(M/\epsilon)^n$ for some constant $C' > 0$; see Proposition 4.2.12 of Vershynin (2018). Since $\epsilon\sqrt{\log(M/\epsilon)} \to 0$ as $\epsilon \to 0^+$, it follows that $\epsilon\sqrt{\log N(\mathcal{A}_{\text{full}}, \epsilon)} \to 0$ as $\epsilon \to 0^+$. $\hfill\square$

## G  PROOF OF THEOREM 10

To derive a matching lower bound, we decompose the regret of Algorithm 1 into that from a reference subset and from missing clusters, using Lemma 16:

**Statement of Theorem 10**  Consider a partition $\mathcal{R} := \{r_\ell\}_{\ell \leq m}$ of the full action space, where each cluster contains more than one action and satisfies Assumption 5. Let $\epsilon := \max_{r \in \mathcal{R}} \text{diam}(r)$. Let $\mathcal{A}$ be the output of Algorithm 1. Then, for the same constant $C$ in Theorem 6 and another constant $c > 0$, for any reference subset $\mathcal{A}'$:

$$
\begin{aligned}
\mathbb{E}_{\theta,\mathcal{A}}[\text{Regret}] \geq &c \cdot \left(\mathbb{E}_q(1 - q(r))^{2K}\right)^{1/2} \cdot \mathbb{E}_\theta\left[\text{Regret of } \mathcal{A}'\right] \\
\geq &c \cdot \left(\mathbb{E}_q(1 - q(r))^{2K}\right)^{1/2} \cdot \left(\min_{r \in \mathcal{R}} \mathbb{E}_\theta\left[\max_{a \in r} \mu_a\right] - C\epsilon\sqrt{\log|\mathcal{R}|}\right).
\end{aligned}
$$

*Proof.*  Choose an arbitrary point $a_\ell \in r_\ell$ as the representative, such that the subset $\mathcal{A}' = \{a_\ell\}_{\ell \leq m}$ forms a reference subset of Definition 1. Fix $\ell$. As in Lemma 12, define a Gaussian process $(Z_a)_{a \in r_\ell}$ by setting $Z_a := \mu_a - \mu_{a_\ell}$, and define random variable $Y_\ell := \sup_{a \in r_\ell} Z_a$. The key idea is that $\text{Regret} \geq Y_\ell$ whenever $r_\ell \cap \mathcal{A} = \emptyset$ and $a^*(\theta) \in r_\ell$, such that

$$
\begin{aligned}
\mathbb{E}\left[\text{Regret} \,\middle|\, r_\ell \cap \mathcal{A} = \emptyset, a^*(\theta) \in r_\ell\right] &= \mathbb{E}\left[\max_{a \in r_\ell} \mu_a - \max_{a' \in \mathcal{A}} \mu_{a'} \,\middle|\, r_\ell \cap \mathcal{A} = \emptyset, a^*(\theta) \in r_\ell\right] \\
&\geq \mathbb{E}\left[\max_{a \in r_\ell} \mu_a - \mu_{a_\ell} \,\middle|\, r_\ell \cap \mathcal{A} = \emptyset, a^*(\theta) \in r_\ell\right] \qquad (18) \\
&= \mathbb{E}\left[Y_\ell \,\middle|\, a^*(\theta) \in r_\ell\right],
\end{aligned}
$$

where the inequality follows from Assumption 5. The last equality follows from $Y_\ell$ is independent from $r_\ell \cap \mathcal{A} = \emptyset$. Further,

$$
\begin{aligned}
\mathbb{E}[\text{Regret}] &= \sum_{r \in \mathcal{R}} \Pr[r \cap \mathcal{A} \neq \emptyset, a^*(\theta) \in r] \cdot \mathbb{E}\left[\text{Regret} \,\middle|\, r \cap \mathcal{A} \neq \emptyset, a^*(\theta) \in r\right] \\
&\quad + \sum_{r \in \mathcal{R}} \Pr[r \cap \mathcal{A} = \emptyset, a^*(\theta) \in r] \cdot \mathbb{E}\left[\text{Regret} \,\middle|\, r \cap \mathcal{A} = \emptyset, a^*(\theta) \in r\right] \\
&\geq \sum_{r \in \mathcal{R}} q(r) \cdot (1 - q(r))^K \cdot \mathbb{E}\left[\text{Regret} \,\middle|\, r \cap \mathcal{A} = \emptyset, a^*(\theta) \in r\right], \\
&\geq \sum_{\ell \leq m} q(r_\ell) \cdot (1 - q(r_\ell))^K \cdot \mathbb{E}\left[Y_\ell \,\middle|\, a^*(\theta) \in r_\ell\right] \\
&\geq c \cdot \left(\mathbb{E}_q(1 - q(r))^{2K}\right)^{1/2} \cdot \left(\sum_{\ell \leq m} q(r_\ell) \cdot \mathbb{E}^2\left[Y_\ell \,\middle|\, a^*(\theta) \in r_\ell\right]\right)^{1/2} \\
&\geq c \cdot \left(\mathbb{E}_q(1 - q(r))^{2K}\right)^{1/2} \cdot \mathbb{E}_\theta\left[\text{Regret of } \mathcal{A}'\right] \\
&\geq c \cdot \left(\mathbb{E}_q(1 - q(r))^{2K}\right)^{1/2} \cdot \left(\min_{r \in \mathcal{R}} \mathbb{E}_\theta\left[\max_{a \in r} \mu_a\right] - C\epsilon\sqrt{\log|\mathcal{R}|}\right)
\end{aligned}
$$

where the equality uses the tower rule. The first inequality holds because the regret is lower bounded by zero when $r \cap \mathcal{A} \neq \emptyset$ and $a^*(\theta) \in r$, and Lemma 16. The second inequality follows from equation 18. The third inequality uses the Pólya-Szegő inequality (Dragomir, 2004), corresponding to a constant $c > 0$:

$$
c := \frac{1}{2}\left(\sqrt{\frac{M_1 M_2}{m_1 m_2}} + \sqrt{\frac{m_1 m_2}{M_1 M_2}}\right),
$$

where $m_1, m_2, M_1, M_2$ are constants such that

$$0 < m_1 \leq (1 - q(r_\ell))^K \leq M_1, \ 0 < m_2 \leq \mathbb{E}\left[Y_\ell \mid a^*(\theta) \in r_\ell\right] \leq M_2, \ \ \forall \ell \leq m.$$

Note that $m_2$ is positive due to the assumption that each cluster contains more than one action. The fourth inequality follows from Jensen's inequality that for a random variable $X$, $\mathbb{E}[X^2] \geq \mathbb{E}[X]^2$, and Lemma 12. Since the reference subset $\mathcal{A}'$ is arbitrary, the lower bound $c \cdot \left(\mathbb{E}_q(1 - q(r))^{2K}\right)^{1/2} \cdot \mathbb{E}_\theta\left[\text{Regret of } \mathcal{A}'\right]$ holds for any reference subset. The last inequality uses Theorem 6. $\qquad \square$

## H   PROOF OF THEOREM 11

**Definition 2.** *The random process $\{\mu_a\}_{a \in \mathcal{S}}$ is a mean-zero sub-Gaussian process if the process $\mathbb{E}\mu_a = 0$ and has the increment condition:*

$$\forall u > 0, \Pr[|\mu_a - \mu_{a'}| \geq u] \leq 2 \exp\left(-\frac{u^2}{2\|a - a'\|_2^2}\right).$$

Let $\gamma_2(\mathcal{S})$ be the Talagrand's chaining functional (Talagrand, 2014) for a set $\mathcal{S} \in \mathbb{R}^n, n \in \mathbb{N} \cup \{+\infty\}$ and Euclidean norm. It is well-known that under the assumption that $\{\mu_a\}_{a \in \mathcal{S}}$ is a Gaussian process, this gives the tightest bound, for a universal constant $L$:

$$\frac{1}{L}\gamma_2(\mathcal{S}) \leq \mathbb{E}\left[\max_{a \in \mathcal{S}} \mu_a\right] \leq L\gamma_2(\mathcal{S}).$$

Within the proof of the upper bound for general sub-Gaussian process, it also derives the deviation bound for the term $\max_{a \in \mathcal{S}} \mu_a$:

**Lemma 18** (Theorem 2.2.22 of Talagrand (2014)). *Let $\{\mu_a\}_{a \in \mathcal{S}}$ be a mean-zero sub-Gaussian process, then there exists a constant $c > 0$:*

$$\Pr\left[\sup_{a, a' \in \mathcal{S}} |\mu_a - \mu_{a'}| \geq u\right] \leq 2 \exp\left(-\frac{cu^2}{\gamma_2^2(\mathcal{S})}\right).$$

**Lemma 19.** *Let $\{\mu_a\}_{a \in \mathcal{A}_{full}}$ be a mean-zero sub-Gaussian process. Consider a partition $\mathcal{R} := \{r_\ell\}_{\ell \leq m}$ of the full action space, and an arbitrary reference subset $\mathcal{A} := \{a_\ell\}_{\ell \leq m}$.*

*Fix $\ell$, define a non-negative random variable*

$$Y_\ell := \sup_{a \in r_\ell} \mu_a - \mu_{a_\ell}.$$

*Then, for some constant $C > 0$,*

$$\mathbb{E}_\theta\left[\max_{\ell \leq m} Y_\ell\right] \leq C\sqrt{\log m} \cdot \max_{\ell \leq m} \gamma_2(r_\ell).$$

*Proof.* Let $\epsilon := \max_{\ell \leq m} \gamma_2(r_\ell)$. For any $\ell \leq m$, we have

$$\Pr\left[|Y_\ell| \geq u\right] = \Pr\left[\left|\sup_{a \in r_\ell} \mu_a - \mu_{a_\ell}\right| \geq u\right]$$

$$\leq \Pr\left[\sup_{a, a' \in r_\ell} |\mu_a - \mu_{a'}| \geq u\right]$$

$$\leq 2 \exp\left(-\frac{cu^2}{\gamma_2^2(r_\ell)}\right)$$

$$\leq 2 \exp\left(-\frac{cu^2}{\epsilon^2}\right),$$

where the equality uses definition of $Y_\ell$. The first inequality uses

$$\left|\sup_{a \in r_\ell} \mu_a - \mu_{a_\ell}\right| \leq \sup_{a \in r_\ell} |\mu_a - \mu_{a_\ell}| \leq \sup_{a, a' \in r_\ell} |\mu_a - \mu_{a'}|.$$

The second inequality uses the fact that $\{\mu_a\}_{a \in r_\ell}$ is a mean-zero sub-Gaussian process and applies Lemma 18. The last inequality uses the definition of $\epsilon$, and $c > 0$ is a constant.

By union bound, we have

$$\Pr\left[\max_{\ell \leq m} |Y_\ell| \geq u\right] \leq 2m \exp\left(-\frac{cu^2}{\epsilon^2}\right).$$

Further, using Lemma 13, we have for some absolute constant $C > 0$:

$$\mathbb{E}\left[\max_{\ell \leq m} |Y_\ell|\right] \leq C\epsilon\sqrt{\log m}.$$

$\square$

**Statement of Theorem 11:** Let $\mathcal{A}$ be output of Algorithm 1. Let $\{\mu_a\}_{a \in \mathcal{A}_{\text{full}}}$ be a mean-zero sub-Gaussian process. Consider a partition $\mathcal{R} := \{r_\ell\}_{\ell \leq m}$ of the full action space. Then, for the same constant $C > 0$,

$$\mathbb{E}_{\theta, \mathcal{A}}[\text{Regret}] \leq C\sqrt{\log m} \cdot \max_{\ell \leq m} \gamma_2(r_\ell)$$

$$+ \left(\mathbb{E}_q\left[(1 - q(r))^{2K}\right] \cdot \mathbb{E}_\theta\left[\max_{a \in \mathcal{A}_{\text{full}}} \mu_a^2\right]\right)^{1/2}.$$

*Proof.* If $r_\ell \cap \mathcal{A} \neq \emptyset$, define $a_\ell \in r_\ell \cap \mathcal{A}$. If $r_\ell \cap \mathcal{A} = \emptyset$, choose an arbitrary point $a_\ell \in r_\ell$ as the representative. The set $\mathcal{A}' := \{a_\ell\}_{\ell \leq m}$ forms a reference subset of Definition 1.

For each $\ell \leq m$, define a random variable

$$Y_\ell := \sup_{a \in r_\ell} \mu_a - \mu_{a_\ell}.$$

Following the same reasoning used in the proof of Theorem 7:

$$\mathbb{E}[\text{Regret}] \leq \mathbb{E}\left[\max_{\ell \leq m} Y_\ell\right] + \left(\mathbb{E}_q\left[(1 - q(r))^{2K}\right] \cdot \mathbb{E}_\theta\left[\max_{a \in \mathcal{A}_{\text{full}}} \mu_a^2\right]\right)^{1/2}$$

$$\leq C\sqrt{\log m} \cdot \max_{\ell \leq m} \gamma_2(r_\ell) + \left(\mathbb{E}_q\left[(1 - q(r))^{2K}\right] \cdot \mathbb{E}_\theta\left[\max_{a \in \mathcal{A}_{\text{full}}} \mu_a^2\right]\right)^{1/2},$$

The first inequality follows from equation 13 in the proof of Theorem 7. The last inequality uses Lemma 19. $\square$

## I  PROPERTIES OF GAUSSIAN WIDTH

Given a set $\mathcal{S} \in \mathbb{R}^n$, the term $\mathbb{E}[\max_{a \in \mathcal{S}} \mu_a]$ where $\theta \sim \mathcal{N}(0, I)$ is called Gaussian (mean) width.

**Claim 1:** $\mathbb{E}\left[\max_{a \in \mathcal{S}}\langle a, \theta\rangle\right] = \mathbb{E}\left[\max_{a' \in -\mathcal{S}}\langle a', \theta\rangle\right].$

$$\mathbb{E}\left[\max_{a \in \mathcal{S}}\langle a, \theta\rangle\right] = \mathbb{E}\left[\max_{a \in \mathcal{S}}\langle a, -\theta\rangle\right] = \mathbb{E}\left[\max_{a \in \mathcal{S}}\langle -a, \theta\rangle\right] = \mathbb{E}\left[\max_{a' \in -\mathcal{S}}\langle a', \theta\rangle\right],$$

where the first equality uses $-\theta$ and $\theta$ are identically distributed. The third equality uses for any $a \in \mathcal{S}$, it holds that $-a \in -\mathcal{S}$.

**Claim 2:** $\mathbb{E}\left[\max_{a \in \mathcal{S}}\langle a, \theta\rangle\right] \leq \frac{1}{2}\mathbb{E}\left[\max_{a, a' \in \mathcal{S}}\langle a - a', \theta\rangle\right].$ Let $a^*(-\mathcal{S}, \theta)$ denote the optimal action in $\mathcal{S}$ for bandit instance $\theta$.

$$2 \cdot \mathbb{E}\left[\max_{a \in \mathcal{S}}\langle a, \theta\rangle\right] = \mathbb{E}\left[\max_{a \in \mathcal{S}}\langle a, \theta\rangle\right] + \mathbb{E}\left[\max_{a' \in -\mathcal{S}}\langle a', \theta\rangle\right]$$

$$= \mathbb{E}\left[\langle a^*(\mathcal{S}, \theta), \theta\rangle\right] + \mathbb{E}\left[\langle a^*(-\mathcal{S}, \theta), \theta\rangle\right]$$

$$= \mathbb{E}\left[\langle a^*(\mathcal{S}, \theta) + a^*(-\mathcal{S}, \theta), \theta\rangle\right]$$

$$\leq \mathbb{E}\left[\max_{a, a' \in \mathcal{S}}\langle a - a', \theta\rangle\right],$$

where the first equality uses Claim 1, the second equality uses the definition of $a^*(-\mathcal{S}, \theta)$. The third equality uses linearity of expectation. The inequality uses that the vector $a^*(\mathcal{S}, \theta) + a^*(-\mathcal{S}, \theta)$ belongs to the set of vectors $\{a - a' : a, a' \in \mathcal{S}\}$. In fact, one can prove the equality that $\mathbb{E}\left[\max_{a \in \mathcal{S}} \langle a, \theta \rangle\right] = \frac{1}{2}\mathbb{E}\left[\max_{a, a' \in \mathcal{S}} \langle a - a', \theta \rangle\right]$, but we only need inequality to prove the next claim.

**Claim 3:** $\mathbb{E}\left[\max_{a \in \mathcal{S}} \mu_a\right] \leq \frac{\text{diam}(\mathcal{S})}{2} \cdot \sqrt{n}$.

$$\mathbb{E}\left[\max_{a \in \mathcal{S}} \mu_a\right] = \mathbb{E}\left[\max_{a \in \mathcal{S}} \langle a, \theta \rangle\right]$$

$$\leq \frac{1}{2}\mathbb{E}\left[\max_{a, a' \in \mathcal{S}} \langle a - a', \theta \rangle\right]$$

$$\leq \frac{1}{2}\mathbb{E}\max_{a, a' \in \mathcal{S}} \|\theta\|_2 \|a - a'\|_2$$

$$\leq \frac{1}{2}\mathbb{E}\,\text{diam}(\mathcal{S})\|\theta\|_2 \leq \frac{\text{diam}(\mathcal{S})}{2} \cdot \sqrt{n},$$

where the first equality uses the definition of $\mu_a$. The first inequality uses Claim 2. The second inequality uses Cauchy-Schwarz inequality. The third inequality uses the definition of $\text{diam}(\cdot)$. The last inequality uses $\mathbb{E}\|\theta\|_2 \leq \sqrt{n}$.

## J  BOUNDS OF EXPECTED MAXIMUM OF GAUSSIAN

Let $X_1, \ldots, X_N$ be $N$ random Gaussian variables (no necessarily independent) with zero mean and variance of marginals smaller than $\sigma^2$, then

$$\mathbb{E}\left[\max_{i=1,\ldots,N} X_i\right] \leq \sigma\sqrt{2\log N}.$$

*Proof.* for any $\delta > 0$,

$$\mathbb{E}\left[\max_{i=1,\ldots,N} X_i\right] = \frac{1}{\delta}\mathbb{E}\left[\log\exp(\delta \max_{i=1,\ldots,N} X_i)\right] \leq \frac{1}{\delta}\log\mathbb{E}\left[\exp(\delta \max_{i=1,\ldots,N} X_i)\right]$$

$$= \frac{1}{\delta}\log\mathbb{E}\left[\max_{i=1,\ldots,N} \exp(\delta X_i)\right] \leq \frac{1}{\delta}\log\sum_{i=1}^{N}\mathbb{E}\left[\exp(\delta X_i)\right]$$

$$\leq \frac{1}{\delta}\log\sum_{i=1}^{N}\exp(\sigma^2\delta^2/2) = \frac{\log N}{\delta} + \frac{\sigma^2\delta}{2},$$

where the first inequality uses Jensen's inequality. Taking $\delta := \sqrt{2(\log N)/\sigma^2}$ yields the results. $\qquad\square$

Let $X_1, \ldots, X_N$ be i.i.d. $\mathcal{N}(0, \sigma^2)$ random variables, then according to (Kamath, 2015):

$$\mathbb{E}_\theta\left[\max_{i=1,\ldots,N} X_i\right] \geq \frac{\sigma\sqrt{\log N}}{\sqrt{\pi\log 2}}.$$

## K  THE EFFECT OF CLUSTERING STRUCTURE ON REGRET

In Figure 3, we study the effect of the cluster diameters (controlled by a spread parameter) on regret. We structure the clustered action space: Five center points are fixed on the unit sphere in $\mathbb{R}^3$, and around each center, 200 points are sampled to form five clusters. Each point is obtained by adding Gaussian noise (mean zero, standard deviation equal to the spread parameter) to the center direction, followed by projection back onto the unit sphere. Bandits are sampled from a 3-dimensional standard Gaussian distribution, i.e., $\theta \sim \mathcal{N}(0, I)$. The left subplot shows the expected regret of Algorithm 1 with $K = 10$, computed using $10^4$ additional bandits, as the spread varies from 0.01 to 0.5. The curves and error shade represent the mean $\pm$ one standard deviation of expected regret over 30 repetitions. The middle and right subplots display example action spaces for spread values of 0.01 and 0.5, with representative actions (purple stars) selected by Algorithm 1 with $K = 10$.

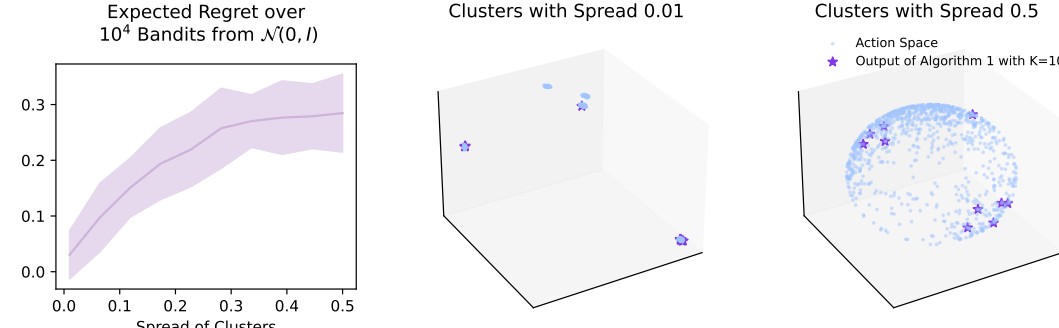

Figure 3: Illustration of clustered action spaces on unit sphere in $\mathbb{R}^3$ and the effect of cluster diameters on regret. Five clusters are formed by generating 5 fixed center points, with 200 points sampled around each using Gaussian noise (spread controls the variance). Bandits are drawn from $\mathcal{N}(0, I)$. The left subplot shows the mean $\pm$ standard deviation of the expected regret (over 30 trials) as the spread varies from 0.01 to 0.5, using $10^4$ additional bandits. The middle and right subplots show example action spaces (blue dots) for spread values 0.01 and 0.5, with representative actions (purple stars) selected by Algorithm 1 with $K = 10$.

## L    VARYING-DEPENDENCE ACTIONS WITH RBF/GIBBS KERNELS

We study the effect of varying action dependence using a Gaussian process with a kernel. To control the degree of dependence, we use stationary RBF kernel and non-stationary Gibbs kernel (Williams & Rasmussen, 2006).

$$
\begin{aligned}
\underset{\text{RBF}}{k}(a, a') &= \exp\left(-\frac{\|a - a'\|^2}{2l^2}\right), \\
\underset{\text{Gibbs}}{k}(a, a') &= \sqrt{\frac{2\,l(a)l(a')}{l(a)^2 + l(a')^2}} \exp\left(-\frac{\|a - a'\|^2}{l(a)^2 + l(a')^2}\right),
\end{aligned}
\tag{19}
$$

where $l$ is a length-scale parameter and $l(a) := 0.1 + 0.9 \cdot \exp(-\|a\|^2)$ is a location-dependent length-scale function. Both of them control the dependence between actions. But, unlike the stationary RBF kernels, the Gibbs kernel allows the correlation to depend not only on the distance between actions, but also on their locations. When $l(a) = l$ is a constant, the Gibbs kernel reduces to the RBF kernel.

**Sampling Outcome Functions from a Kernel:** We first construct the kernel matrix $\mathbf{K}$, where each entry is given by $\mathbf{K}_{a,a'} = k(a, a')$, for $a, a' \in \mathcal{A}_{\text{full}}$, depending on the choice of kernel. We then sample a Gaussian vector (a Gaussian process function evaluated at finite input) $f \sim \mathcal{N}(0, \mathbf{K})$. Under either kernels defined in equation 19, the variance of the function value $f(a)$ is one for all $a \in \mathcal{A}_{\text{full}}$. In this way, we sample functions from a RKHS function class; See (Kanagawa et al., 2018, Theorem 4.12).

To study the effect of varying dependence on the output of Algorithm 1, we consider a fixed action space consisting of 1000 grid points in the interval $[0, 2]$, using Gibbs kernel in equation 19 to sample outcome functions. To simplify computations, we marginalize a Gaussian process defined by the kernel over the grid. Figure 4 provides examples of sampled outcome functions (blue curves, with the y-axis on the right-hand side), which become smoother as the actions approach the left end of the interval—indicating stronger correlations among function values in that region. We run Algorithm 1 on this action space with $K = 5000$ to select actions and record the frequency of each action being selected. The resulting histogram (purple bars, with the y-axis on the left) reflects the importance measure $q$, highlighting that Algorithm 1 tends to select more actions from regions where the outcome functions are rougher—i.e., where action outcomes are less correlated and their features $\Phi(a)$ are are farther apart. Another interesting aspect of this subplot is the two high bars at the edges. Recall that the actual action space consists of feature vectors $\Phi(a)$ for $a \in \mathcal{A}_{ful}$. For actions indexed closer to 0, their feature vectors become more densely packed compared to those

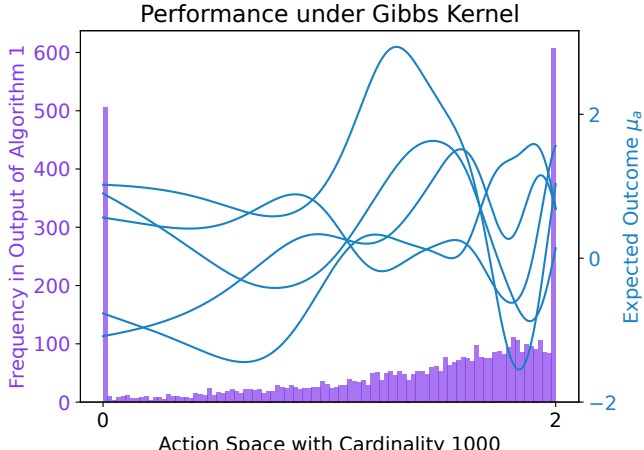

Figure 4: Experiments with outcome functions sampled from RBF/Gibbs kernels in equation 19. Sampled outcome functions from Gibbs kernel over fixed 1000 grid points in $[0, 2]$ (blue curves, right y-axis). The histogram (purple bars, left y-axis) shows action selection frequencies by Algorithm 1 with $K = 5000$, favoring regions with rougher functions and edge points.

indexed closer to 2, resulting in more correlated outcomes. The two actions at the edges, indexed by 0 and 2, correspond to the two farthest points in the actual feature space.

## M INDEPENDENT AND IDENTICALLY DISTRIBUTED ACTIONS

As an extreme case, suppose that $\{\mu_a\}_{a \in \mathcal{A}_{\text{full}}}$ is a set of i.i.d. random variables. This corresponds to the canonical process in which the action space is given by the orthonormal basis of $\mathbb{R}^n$, where $n = |\mathcal{A}_{\text{full}}|$. To see this, we associate each action $a$ with a unit vector $e_a$, which has a value of 1 at the $a$th coordinate and 0 elsewhere, and define the expected outcome as $\mu_a(\theta) := \langle e_a, \theta \rangle$. With this construction, the collection $\{\mu_a\}_{a \in \mathcal{A}_{\text{full}}}$ consists of mutually independent random variables.

Let the set $\mathcal{A}_{\text{full}} = \{e_i : i = 1, \ldots, n\}$ denote the $n$ unit vectors aligned with the coordinate axes in $\mathbb{R}^n$. Hence, $\text{diam}(\mathcal{A}_{\text{full}}) = \sqrt{2}$. In this case, $\max_{a \in \mathcal{A}_{\text{full}}} \langle a, \theta \rangle$ is equivalent to the maximum among the $n$ entries of $\theta$, where each entry is i.i.d. from the standard normal distribution $\mathcal{N}(0, 1)$. By symmetry, each coordinate has the same probability of attaining the maximum value. If each cluster contains only one unit vector, then the importance measure $q$ over clusters is the uniform distribution.

**Expected maximum in $\mathcal{A}_{\text{full}}$:** Let $X_i$, $i = 1, \ldots, n$ be i.i.d. samples from $\mathcal{N}(0, 1)$, then

$$\mathbb{E} \max_{a \in \mathcal{A}_{\text{full}}} \mu_a = \mathbb{E}_\theta \max_{i=1,\ldots,n} \theta_i = \mathbb{E} \max_{i=1,\ldots,n} X_i,$$

where the equivalence comes from that each entry $\theta_i$ is i.i.d. sample from $\mathcal{N}(0, 1)$. Then, according to the bounds of expected maximum of Gaussian in Supplementary J, we have

$$\frac{\sqrt{\log n}}{\sqrt{\pi \log 2}} \leq \mathbb{E} \max_{a \in \mathcal{A}_{\text{full}}} \mu_a \leq \sqrt{2 \log n}, \quad \frac{\sqrt{\log |\mathcal{A}|}}{\sqrt{\pi \log 2}} \leq \mathbb{E} \max_{a \in \mathcal{A}} \mu_a \leq \sqrt{2 \log |\mathcal{A}|}.$$

**Bounds of expected regret of arbitrary $\mathcal{A}$:** By definition of regret in equation 1,

$$\frac{\sqrt{\log n}}{\sqrt{\pi \log 2}} - \sqrt{2 \log |\mathcal{A}|} \leq \mathbb{E}_\theta[\text{Regret}] \leq \sqrt{2 \log n} - \frac{\sqrt{\log |\mathcal{A}|}}{\sqrt{\pi \log 2}}.$$

As a result, any algorithm, including Algorithm 1, would perform poorly unless the subset size $|\mathcal{A}|$ is sufficiently large.

