# OpenReview forum: "Representative Action Selection for Large Action Space Meta-Bandits"
_ICLR.cc/2026/Conference — Submitted to ICLR 2026_

### Official Review · Reviewer_jxrm · 2025-10-22

**Soundness:** 3
**Presentation:** 2
**Contribution:** 2
**Rating:** 6
**Confidence:** 2

**Summary:**

This work tackles selecting a subset of actions from a large shared action space in bandit problems, aiming to achieve performance close to using the full set. It models action similarity with a Gaussian process and proposes a simple ε-net algorithm to choose representative actions. It provides theoretical guarantees and empirically compare the method to Thompson Sampling and UCB, showing competitive performance.

**Strengths:**

- The authors provide lower and upper bounds, which makes the proposed algorithm more convincing.
- Numerical experiments are provided, which helps readers to understand the charcteristics of the algorithm.

**Weaknesses:**

- Although some related works are mentioned, the authors do not clearly explain how they are connected to the problem setting.
- What is the main takeaway from the theoretical analysis? Is it primarily about the relationship between $\epsilon$ and the regret?
- Section 4 is overly long and difficult to follow.

**Questions:**

- I could not understand the main message of Section 3. What is the main message here?
- If we separate Section 4 into some subsections, what whould they be?

---

### Official Review · Reviewer_4i2i · 2025-11-01

**Soundness:** 2
**Presentation:** 2
**Contribution:** 2
**Rating:** 2
**Confidence:** 3

**Summary:**

This paper introduces a framework for selecting a representative subset from a large action space shared by a family of bandits, trying to achieve performance close to that of using the full action space. The authors propose an ε-net–based algorithm and provide theoretical performance guarantees, as well as an empirical comparison to Thompson Sampling and UCB.

**Strengths:**

- The authors  try to bring high dimensional geomtery (epsilon nets) to bandits with large action spaces
- They provide a regret analysis of their proposed algorithm.

**Weaknesses:**

The paper starts from a motivation about handling large action spaces shared y a family of bandits, but it is not concretely instantiated for any specific family of bandits. It is formulated as an abstract framework whose significance is hard to understand without concrete instantiation in specific bandit families. the nature of the work seems more suited to a learning theory conference like COLT rather than to ICLR.

Several details are also very unclear. Algorithm 1 seems extremely general and trivial and I don't see how it relates to the claim on lines 249-252 that the algorithm samples according to the importance measure q.

The paper is motivated by large action spaces but the experiments deal with extremely small toy action spaces - pretty clear that this experimental part was an add on, and doesn't make the case for large action spaces. Also in this simple example, I don't see which family of bandits is being referred to, and there is a lot of standard general stuff about RKHS that does not seem very relevant.

**Questions:**

- How is Algorithm 1 magically sampling from the importance measure q? It doesn't appear at all in the description of the algorithm
-  Could you give a specific example of families of bandits for which your results give something new?
- How does your work compare to two well known approaches to large action spaces, the X-armed bandits of Bubeck et al and the Zooming algorithm of Kleinberg et al?

---

### Official Review · Reviewer_Vmxp · 2025-11-03

**Soundness:** 1
**Presentation:** 2
**Contribution:** 1
**Rating:** 2
**Confidence:** 3

**Summary:**

This paper studies choosing a representative subset of actions from the original action space in linear bandit problems. More formally, it would like to choose the subset such that the optimal expected reward in the subset approximates the globally optimal expected reward, with performance quantitatively characterized by the regret defined in Eq. (1). It proposes Alg. 1 and proves some theoretical guarantees of it. Experiments show that the algorithm outperforms other baselines such as successive halving, combinatorial Thompson sampling, and combinatorial UCB.

**Strengths:**

Contextual bandits with large action spaces has many applications, and the attempts to reduce the action space to mitigate the computational cost is meaningful.

**Weaknesses:**

- From a statistical perspective, reducing the size of the action space in linear bandits does not provide a lot of improvement in the bandit regret. Importantly, the regret for linear bandits depends usually on the linear dimension of the action space, which is much smaller than the cardinality of the action space (e.g. "Bandit Algorithms", Theorem 36.4). In this respect I think the motivation in lines 58-75 (using the cardinality of the action space as the main argument) is somewhat flawed.

- For Theorems 6 and 7 I am worried that the first term max_{r in R} E_\theta[ \max_{a \in r} \mu_a ] is not vanishing when eps -> 0? Why is this a meaningful result?

- For Theorem 9 I am afraid that we cannot escape the curse of dimensionality -- epsilon needs to be smaller than a constant for it to be useful? But then N(A_full, eps) is usually exponential in n (the dimension of the action space), which makes the required K also exponential in n?

- I am also not buying the application background of the problem setup. It looks like one is using meta learning to learn a representative subset of the action space. But this paper assumes that one knows the exact prior of the task parameter \theta? Is this prior learned by interaction with the previous tasks? If so, I am not sure if such prior can be assumed to be known exactly, since after learning in historical tasks, we may not learn the ground truth task parameter for those tasks, in a pointwise manner.

**Questions:**

See questions above. Also:

- CTS and CUCB, in my understanding, works when the reward function is additive across different arms in a super-arm. But here the reward function of a super arm is the maximum of reward of arms therein. Thus, I am not sure if this is a fair comparison..

---

### Author Response · Authors · 2025-11-13
**Rebuttal: To Reviewer Vmxp**

We thank the reviewer for their feedback, but we believe there was a considerable missunderstanding of the paper.

> reducing the size of the action space in linear bandits does not provide a lot of improvement in the bandit regret

Our framework is not restricted to linear bandits. We state that it applies to sub-Gaussian processes in the main body, and the most general version is in line 643 of the Appendix, which does not require the linear structure of *a* and \theta.

Even under the Gaussian process assumption in Eq. 3, our framework assumes the feature vector of action *a* is unknown to the decision-maker. This practical assumption contrasts with linear bandits. In lines 104-111, we mention that linear bandits assume the feature vector is known, making the action space dimension irrelevant. This is also reflected in the book "Bandit Algorithms": linear bandits are introduced in Chapter 19, where the text around Eq. 19.1 states that "the learner has access to a map \psi... for an unknown parameter vector \theta_*" and "\psi is called a feature map."

> For Theorems 6 and 7 I am worried that the first term max_{r in R} E_\theta[ \max_{a \in r} \mu_a ] is not vanishing when eps -> 0? Why is this a meaningful result?

Theorem 9 shows the regret bound in Theorems 6 and 7 goes to zero as K increases. For the first term, our assumption of a canonical Gaussian process means for each cluster r, the term E_\theta[ \max_{a \in r} \mu_a ] is bounded between \frac{1}{\sqrt{2\pi}} * \text{diam}(r) and \frac{\sqrt{n}}{2} * \text{diam}(r) (a standard result in high-dimensional probability; see Proposition 7.5.2 (f) of (Vershynin, 2018)).

Thus, as epsilon goes to zero, the diameter of r goes to zero, and the first term follows.

> For Theorem 9 I am afraid that we cannot escape the curse of dimensionality -- epsilon needs to be smaller than a constant for it to be useful? But then N(A_full, eps) is usually exponential in n (the dimension of the action space), which makes the required K also exponential in n?

The measure q is key to our results. If q is highly concentrated, we need far fewer actions to cover the important regions. If q is uniform, we must form a geometric epsilon-net over the action space, whose smallest cardinality is the covering number and can scale with dimension n. Theorem 9 shows that even in the worst case where q is uniform, the expected regret still goes to zero. This is why the covering number N(A_full, eps) appears in the results.

Regarding the curse of dimensionality: for an action space of [0,1]^n, we need about (1/ε)^n points for universal coverage. However, if the support of q is the main diagonal {(t,t,…,t) : t∈[0,1]}, we need at most 1/ε points to achieve the same expected regret as universal coverage. Please let us know if this is clear and helpful.

> this paper assumes that one knows the exact prior of the task parameter \theta? Is this prior learned by interaction with the previous tasks?

We do not assume the distribution of task parameters is known, but we require the ability to sample tasks (bandit instances).

> CTS and CUCB works when the reward function is additive across different arms in a super-arm. But here the reward function of a super arm is the maximum of reward of arms therein. Thus, I am not sure if this is a fair comparison..

In lines 146-153, we discussed that CTS and CUCB are not designed for our framework. However, they are the closest applicable baselines for combinatorial problems with large action spaces.

The experiments in Figure 1 are more reasonable, but using TS, UCB, and Successive Halving at the super-arm level suffers from combinatorial explosion and is only feasible in small action spaces.

---

### Author Response · Authors · 2025-11-13
**Rebuttal: To Reviewer 4i2i**

> It is formulated as an abstract framework whose significance is hard to understand without concrete instantiation in specific bandit families.

The specific family of bandits is defined in Eq. 3. This family is generalized to sub-Gaussian processes in Appendix B. In line 45, we state that each bandit is a contextual bandit, referencing (Dani et al., 2008).

> Algorithm 1 seems extremely general and trivial

Algorithm 1 is indeed simple. However, the point of our work is that we can exploit correlations among action space and use it to tackle a combinatorial problem. We believe it is not so trivial that the simple algorithm does this and it also works well.

> I don't see how it relates to the claim on lines 249-252 that the algorithm samples according to the importance measure q.

The algorithm repeats the following process K times: sample a bandit environment and find its optimal action. The measure q is the probability that a region of the action space contains the optimal action over the distribution of bandits.

Recall the medicine example: the algorithm essentially asks random customers which drug they want and stocks that drug. This makes it likely that we will keep high-demand drugs in stock—the ones many customers want to buy (high q measure).

> The paper is motivated by large action spaces but the experiments deal with extremely small toy action spaces

Choosing 10 actions from 500 is a combinatorial problem with 2.458×10^20 super-arms.

> Also in this simple example, I don't see which family of bandits is being referred to, and there is a lot of standard general stuff about RKHS that does not seem very relevant.

We mention RKHS because if outcome functions are sampled from it, the resulting bandit family is of the form in line 367, aligning with our formulation in Eq. 3 and allowing our algorithm to be applied.

By standard results in Williams & Rasmussen, (2006), the Gaussian kernel admits an (infinite-dimensional) feature map \Phi. In our experiments, \Phi is unknown to the decision-maker. The set \mathcal{A}_{full} serves only as an index set (e.g., action names), while the true action space is the image of the action space under \Phi.

> Could you give a specific example of families of bandits for which your results give something new? How does your work compare to two well known approaches to large action spaces, the X-armed bandits of Bubeck et al and the Zooming algorithm of Kleinberg et al?

X-armed bandits and the Zooming algorithm work on a single bandit environment. We work with a family of bandits where the mean-payoff function differs in each instance. We also do not assume a known metric structure on the actions.

Our objective is also different: we assume individual bandits can be solved (by an oracle, and we only need to solve a few instances). Instead, we focus on the combinatorial problem of finding a small subset of actions that effectively represents the large action space.

Our contribution is a new framework where a large action space is shared across a family of bandits, with actions’ mean rewards correlated in each instance but the similarity structure is unknown to the decision-maker. We formulate this using a Gaussian process, extended to sub-Gaussian distributions. A further contribution is the new problem of finding a fixed action subset to represent the large space, for which we provide an efficient algorithm.

---

### Author Response · Authors · 2025-11-13
**Rebuttal: To Reviewer jxrm**

> Although some related works are mentioned, the authors do not clearly explain how they are connected to the problem setting.

Optimal Action Identification, Stochastic Linear Optimization, and GP Optimization relate to Step 6 of our algorithm, which requires finding an optimal action in a sampled bandit environment.

Top-K Action Identification and Combinatorial Bandits relate to our core problem of selecting a representative action subset from a large space, serving as our closest baselines.

Please indicate the specific parts of related work so we can clarify and improve.

> What is the main takeaway from the theoretical analysis? Is it primarily about the relationship between epsilon and the regret?

We show that correlations among actions can be used, so an action can be represented by its neighbors. Theorems 6-7 show our algorithm performs comparably to universal coverage of the action space. The key point in line 323 is the price we pay for not using universal coverage; it depends on the number of samples K and the importance measure q.

This price is highest when q is uniform. Theorem 9 shows the expected regret of our algorithm still goes to zero, even in this worst case.

The relationship between epsilon and regret is also tested via a 3-dimensional example in Figure 3 in Appendx.

> If we separate Section 4 into some subsections, what whould they be?

Section 4 details the experimental setup and presents two experiments.

First, we explain how RKHS functions connect to our framework in Eq. 3, which justifies sampling them for our experiments.

The first experiment compares our method against using TS, UCB, and Successive Halving at the super-arm level.

The second experiment compares our method against CTS and CUCB at the base-arm level.

---

### Meta-Review · Area_Chair_JkwN · 2025-12-26

**Summary:**

This paper addresses the challenge of selecting a representative subset of actions from a large action space in multi-armed bandits. It posits that similar actions exhibit correlated payoffs, which are modeled using a Gaussian process. The authors introduce an $\epsilon$-net algorithm designed to choose a subset of actions that can achieve near-optimal performance without having to utilize the entire action space. The paper includes theoretical guarantees regarding the performance of the proposed method and presents empirical comparisons with established methods like Thompson Sampling (TS) and Upper Confidence Bound (UCB) in various benchmarks.

The most important reviewers' concerns include:

Theoretical Clarity: Many reviewers felt that the theoretical contributions were difficult to follow and understand, particularly concerning the assumptions related to Gaussian processes and how they connect to the algorithm's performance.

Practical Applicability: Reviewers highlighted that the experiments were conducted using relatively small toy examples, which seemed disconnected from the intended application in large action spaces. The simplicity of the examples raised concerns about the algorithm's applicability to more complex, real-world scenarios.

Algorithmic Generalization: Some reviewers pointed out that the presented Algorithm 1 appeared overly general and trivial, lacking clear mechanisms for how it samples based on the importance measure q.

Relation to Existing Work: There was confusion regarding how the proposed method related to known approaches for dealing with large action spaces, particularly questioning how it compared to X-armed bandits and the Zooming algorithm.

Empirical Validation: Reviewers expressed a desire for more empirical validation, especially results involving larger models and more diverse datasets to substantiate the claims made about the efficacy of the algorithm.

**Reviewer Concerns:**

I agree that the work lacks some clarity for its novelty to be appreciated.

For example, in the algorithm, it is not at all clear how the algorithm samples according to the importance measure q. The interpretation of Theorems 6 and 7 could also be enhanced. I agree with the reviewers that it's not clear how the first term behaves. The most important comment, perhaps, pertains to why we want to study linear bandits with large action spaces $K$ when it is well known that the regret scales largely with $d$, the dimension of the feature vectors, rather than $K$ the number of actions.

I believe moving forward, it would be advisable for the authors to better motivate the work by framing it primarily as a online combinatorial optimization problem rather than from the "large action space" perspective.

**Reviewer Scores:**

Unfortunately, the reviewers did not engage with the authors during the author-reviewer discussion period, so I cannot infer how they would have changed their scores.

---

### Decision · Program_Chairs · 2026-01-26

Reject